# Deactivation of redox mediators in lithium-oxygen batteries by singlet oxygen

Won-Jin Kwak [1], Hun Kim [1], Yann K. Petit[2], Christian Leypold[2], Trung Thien Nguyen [1], Nika Mahne[2], Paul Redfern[3], Larry A. Curtiss[3], Hun-Gi Jung [4], Sergey M. Borisov[5], Stefan A. Freunberger [2] & Yang-Kook Sun [1]

Non-aqueous lithium-oxygen batteries cycle by forming lithium peroxide during discharge and oxidizing it during recharge. The significant problem of oxidizing the solid insulating lithium peroxide can greatly be facilitated by incorporating redox mediators that shuttle electron-holes between the porous substrate and lithium peroxide. Redox mediator stability is thus key for energy efficiency, reversibility, and cycle life. However, the gradual deactivation of redox mediators during repeated cycling has not conclusively been explained. Here, we show that organic redox mediators are predominantly decomposed by singlet oxygen that forms during cycling. Their reaction with superoxide, previously assumed to mainly trigger their degradation, peroxide, and dioxygen, is orders of magnitude slower in comparison. The reduced form of the mediator is markedly more reactive towards singlet oxygen than the oxidized form, from which we derive reaction mechanisms supported by density functional theory calculations. Redox mediators must thus be designed for stability against singlet oxygen.

[1] Department of Energy Engineering, Hanyang University, Seoul 04763, Republic of Korea. [2] Institute for Chemistry and Technology of Materials, Graz University of Technology, Graz 8010, Austria. [3] Materials Science Division, Argonne National Laboratory, Illinois 60439, USA. [4] Center for Energy Convergence Research, Green City Technology Institute, Korea Institute of Science and Technology, Seoul 02792, Republic of Korea. [5] Institute for Analytical Chemistry and Food Chemistry, Graz University of Technology, Graz 8010, Austria. Correspondence and requests for materials should be addressed to S.A.F. (email: freunberger@tugraz.at) or to Y.-K.S. (email: yksun@hanyang.ac.kr)

L
ithium-oxygen (Li-O$_2$) batteries have a very high theoretical capacity, but are still far from practical use[1,2]. Among the many problems associated with Li-O$_2$ batteries, the most highlighted issues are their high charge overpotential and side reactions[3–8]. The high charge overpotential due to the difficulty of decomposing the discharge product, lithium peroxide (Li$_2$O$_2$), severely increases side reactions that decompose the electrolyte and electrode and lead to poor rechargeability, increasing charging overpotential, and a build-up of parasitic products during cycling[9–12].

To mitigate the high overpotentials and associated side reactions, catalysts have been utilized to facilitate Li$_2$O$_2$ oxidation during recharging[7,11–14]. The many reported catalysts for decomposing Li$_2$O$_2$ can be classified into two types. The first type, solid catalysts, may enhance Li$_2$O$_2$ decomposition by enhancing charge transport within Li$_2$O$_2$[12,15] or delithiation kinetics[13,16]. However, solid catalysts act only near their surface and may not only accelerate the decomposition of Li$_2$O$_2$ but also the undesired side reactions involving the electrode and electrolyte[12,14,17] The second type, redox mediators (RMs), are soluble catalysts in the electrolyte to chemically decompose Li$_2$O$_2$. They are oxidized at the porous electrode substrate and then diffuse to Li$_2$O$_2$, which decomposes to Li$^+$ and O$_2$ by reforming the original reduced state[18–22]. In principle, mediators with a redox potential beyond the thermodynamic potential of the O$_2$/Li$_2$O$_2$ couple (2.96 V vs. Li/Li$^+$) allow the cell to be recharged with nearly zero overpotential[18–23]. Therefore, many different redox mediators have recently been studied, with a focus on finding the lowest possible voltage and fastest kinetics[7,18–28]. They have been shown to enable recharging at a potential close to their redox potential at rates far greater than those that can be achieved without a mediator.

However, the catalytic effect of RMs deteriorates with repeated cycling. Reported reasons for this deterioration include side reactions with the anode when unprotected lithium metal is used[29–32] and reaction with the electrolyte[18,21,22]. However, even when both of these effects are excluded by protecting the anode with, e.g., a solid electrolyte and choosing mediators that are inert towards the electrolyte, the RMs still gradually degrade and the energy efficiency decreases. To solve these problems, the SOMO energy of the oxidized mediator must not be lower than the HOMO of the electrolyte solvent[22]. However, even when lithium metal and the selected RM were completely separated, the catalytic activity of the RM still declined. This implies that the RM must participate in other side reactions with reactive species at the cathode, which have not yet been clarified[21,33]. Meanwhile, it is well established that reactive oxygen species cause electrolyte decomposition. Superoxide (O$_2^-$) and Li$_2$O$_2$ have traditionally been assumed to cause the majority of side reactions due to their nucleophilicity, basicity, and/or radical nature, even though theoretical calculations suggest unfavorable reaction energies[1,34–40]. Only recently has it been demonstrated that singlet oxygen ($^1$O$_2$), the first excited state of ground state triplet oxygen, is actually the main cause of parasitic reactions during the cycling of metal-O$_2$ batteries[41–45].

Here we assess the reactivity of organic RM's towards dissolved oxygen (O$_2$), potassium superoxide (KO$_2$), Li$_2$O$_2$, and $^1$O$_2$ using quantitative UV–Vis analysis and $^1$H-NMR. We demonstrate the predominant cause for RM deactivation to be $^1$O$_2$. Reactions with the other oxygen species are, if at all detectable, comparatively negligible. The reduced state of the RMs is markedly more reactive than the oxidized state due to the electrophilic nature of $^1$O$_2$. The deactivation mechanisms can therefore be proposed to involve "ene" and "diene" cycloadditions and oxidation of the sulfur; these mechanisms are supported by density functional theory (DFT) calculations and analysis of decomposition

products. The obtained reaction energies agree well with the observed kinetics. Only by clearly identifying the reason for the deactivation of RMs, a key active material in Li-O$_2$ batteries, can more reversible and highly efficient Li-O$_2$ batteries be achieved. Their side reactions with cell components caused by $^1$O$_2$ thus need to be considered comprehensively.

## Results

**Spectroscopic proof for mediator deactivation by $^1$O$_2$.** Redox mediators may be deactivated by any of the potentially reactive species that appear during cycling of the cell, including O$_2$, O$_2^-$, Li$_2$O$_2$, and, as recently revealed, the highly reactive $^1$O$_2$. We thus investigated the stability of a selection of redox mediators towards these species. Among the many kinds of RMs studied so far, we chose tetrathiafulvalene (TTF) and dimethylphenazine (DMPZ) as representative redox mediators, since TTF was amongst the first RMs reported and DMPZ has one of the lowest charge potentials reported and has been shown to be compatible with glyme electrolyte as it does not facilitate its oxidative decomposition based on DFT calculations and experimental verification[22].

The RMs were dissolved in tetraethylene glycol dimethyl ether (TEGDME) at a concentration (60 µM) suitable for UV–Vis spectroscopy. After measuring the fresh solutions, the mediators in solution were then exposed to O$_2$, KO$_2$, Li$_2$O$_2$, and $^1$O$_2$ (Fig. 1 and Supplementary Fig. 1). For the first three, the contact time was 24 h, after which the electrolyte was reexamined. In the case of KO$_2$, an excess amount of 18-crown-6 was added to dissolve the KO$_2$ and thus enhance its reactivity. Dissolved oxygen (O$_2$), O$_2^-$, and Li$_2$O$_2$ had no appreciable effect on the DMPZ or TTF concentration even after 24 h (Fig. 1a–e and Supplementary Fig. 1). Additionally, the NMR spectra of the electrolyte solutions after this time show negligible changes (Supplementary Fig. 2). These results are consistent with those of a previous study that reported the stability of DMPZ against O$_2^-$ (see ref. [22]).

To investigate the stability of the RMs against $^1$O$_2$, we produced $^1$O$_2$ photochemically by illuminating O$_2$-saturated mediator solutions containing 1 µM of the photosensitizer palladium(II) *meso*-tetra(4-fluorophenyl)-tetrabenzoporphyrin (Pd$_4$F) at a wavelength of 643 nm[46]. Photosensitization transfers energy from absorbed light to triplet oxygen[47]. The process is initiated by the excitation of the photosensitizer from its S$_0$ ground state to its excited singlet state S$_n$, which then relaxes to the lowest excited singlet state S$_1$ and yields the triplet state T$_1$ via intersystem crossing (ISC). T$_1$ then transfers the energy to $^3$O$_2$ to form $^1$O$_2$. Unlike O$_2$, KO$_2$, and Li$_2$O$_2$, we found that $^1$O$_2$ decreased the main absorbance peaks of the RMs within several hours for DMPZ and within seconds for TTF (Fig. 1c–f). Simultaneously, new products appeared with red-shifted absorption bands. $^1$H-NMR analyses of the solutions after illumination show the corresponding disappearance of the mediators, Supplementary Fig. 3. The NMR intensity of DMPZ decreased by ~40%, in agreement with the similar decrease observed in its UV absorbance (Fig. 1c). For TTF, the NMR spectra (Supplementary Fig. 3b) show the concurrent appearance of new products. Of note, the integral of all the new products are much less than the mediators at the start. This means that the products still visible in the NMR do not represent all products the mediators are decomposing to. They may form inorganic products or evolve as gases as ultimate products of oxidative decomposition reactions discussed later. Together, these data demonstrate the drastically higher reactivity of the RMs towards $^1$O$_2$ than any of the other oxygen species.

Figure 2 shows the evolution of the main absorbances of the RMs with time when in contact with $^1$O$_2$. $^1$O$_2$ caused DMPZ to

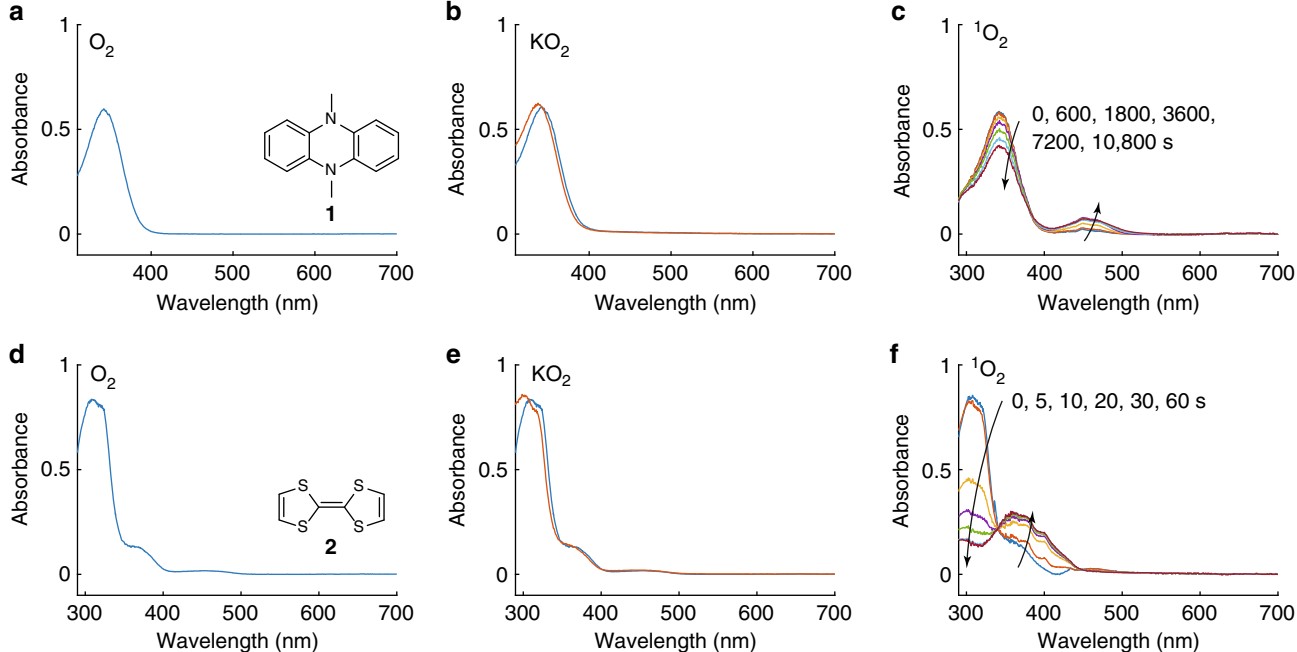

**Fig. 1** Stability of reduced redox mediators against oxygen species. UV–Vis spectra of DMPZ (**a**–**c**) and TTF (**d**–**f**) in 0.1 M LiTFSI/TEGDME electrolyte before and after exposure to $O_2$ (**a**, **d**), $KO_2$ (**b**, **e**), and $^1O_2$ (**c**, **f**). The concentrations of DMPZ and TTF were 60 μM each in their respective solutions. $O_2$ and $KO_2$ (together with an excess of 18-crown-6) were kept in contact with the RMs for 24 h. $^1O_2$ was photogenerated in the $O_2$-saturated solution using 1 μM palladium(II) meso-tetra(4-fluorophenyl)-tetrabenzoporphyrin and illumination at 643 nm, and the spectra were measured after the illumination times indicated. The spectrum of the sensitizer has been subtracted from **c** and **f**

decay to two-thirds of its initial value within 3 h and TTF to be fully decomposed within about half a minute. TTF thus degrades much more rapidly than DMPZ. Although we do not completely rule out the reactivity of the RMs with $O_2$, $KO_2$, or $Li_2O_2$, it is clear from Fig. 1 that the RMs are much more reactive with $^1O_2$. Overall, the data demonstrate that RM deactivation is overwhelmingly associated with $^1O_2$, which has been shown to be formed during both the discharging and charging of the cell[41,43,44].

**Reactivity between $^1O_2$ and oxidized redox mediators**. Singlet oxygen ($^1O_2$) is known to react with electron-rich organic substrates containing C=C double bonds via so-called "ene" or "diene" reactions driven by the electrophilic nature of $^1O_2$[48–51]. The presence of ene and diene motifs in TTF and DMPZ, respectively, makes these mechanisms likely routes of attack leading to RM decomposition, as examined later. Given the electrophilic nature of $^1O_2$, the question arises whether the oxidized forms of the mediators would show similarly strong reactivity. To test this, we oxidized the mediators electrochemically (see Methods and Supplementary Fig. 6 for details), exposed them to in situ generated $^1O_2$ as before, and followed the mediator concentration using UV–Vis (Fig. 3). Considering first DMPZ+, the spectra show a gradual decrease of the main peaks at ~370 and 450 nm together with increasing absorbance at around 400 and 500 nm. This indicates the gradual decomposition of DMPZ+, albeit at a markedly lower rate than DMPZ (Fig. 1c), accompanied by the formation of new products. After 3 h of illumination, >95% of the initial DMPZ+ remained, as compared to only ~65% of the DMPZ at the same time point. The relatively slower reactivity of DMPZ+ compared to DMPZ can equally be seen by comparing Fig. 2a with 3b. However, regardless of the relative stability of DMPZ+ in the presence of $^1O_2$, it is clear that the DMPZ/DMPZ+ redox couple is degraded as a whole by $^1O_2$, because DMPZ reacts strongly.

Turning to TTF+, the analogous experiment is shown in Fig. 3c, d. The spectra do not simply scale over the full wavelength range, but instead show a rapid decrease around 300 nm and a more gradual decrease elsewhere, which indicates the degradation of TTF+ together with the formation of new products. TTF+ is degraded to about two-thirds of its initial concentration within 10 min, whereas TTF was already fully decomposed after only ~30 s (Fig. 1c). The rate of TTF+ decomposition is roughly 150 times lower than that of TTF. As with the DMPZ/DMPZ+ couple, the reduced form TTF reacts much faster than the oxidized form TTF+. Overall, TTF/TTF+ reacts much faster with $^1O_2$ than DMPZ/DMPZ+ regardless of the oxidation state. In all cases, the reaction with $^1O_2$ clearly dominates the possible reactions with the other oxygen species ($O_2$, $O_2^-$, and $Li_2O_2$).

The marked difference in reactivity towards $^1O_2$ between the reduced and oxidized forms points to reaction mechanisms that are governed by the electron-richness of the substrate. The literature on the reactivity of $^1O_2$ with organic substrates most commonly indicates an organic peroxide material as an initial product[52]. Thus, $^1O_2$ produces R–OOH, R•, and R–OO• moieties; these radicals can propagate to generate other reactive intermediates and various by-products, particularly in the presence of $O_2$[53,54]. Therefore, the RMs will lose their redox activity after initial reaction with $^1O_2$ due to the instability of the initial products.

**Mechanisms and energetics of reactions between RM and $^1O_2$**. In Figs. 4, 5, we propose reactions of DMPZ/DMPZ+ and TTF/TTF+, respectively, with $^1O_2$ based on published mechanisms for the reaction of $^1O_2$ with dienes, enes, and sulfides, respectively[48–51,55–57]. We also carried out DFT calculations of the free energies of these reactions and calculated barriers for the reactions by finding transition states to judge their likelihood. The DFT energies of neutrals and cations were calculated at the B3LYP/6–31+G* level of theory[58] for their geometries optimized at the

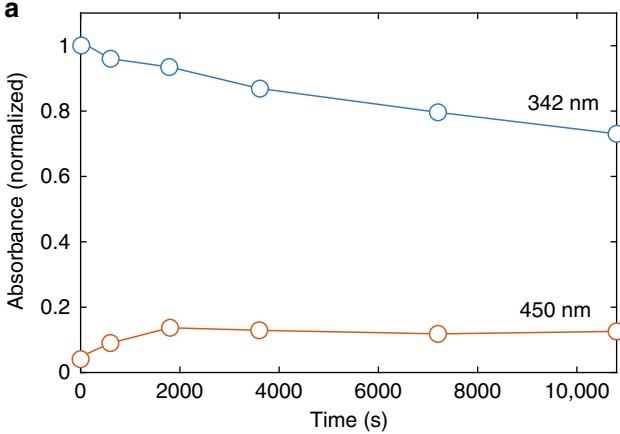

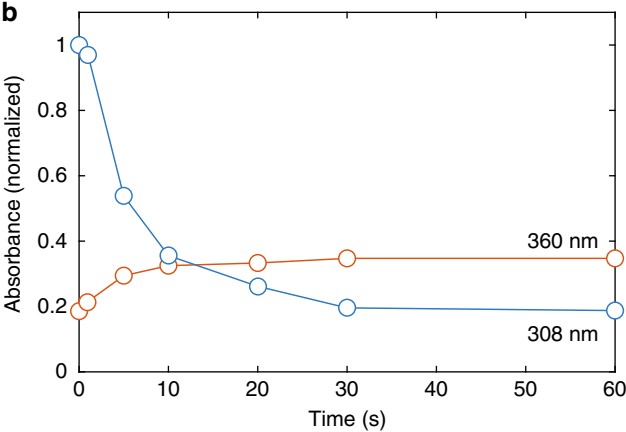

**Fig. 2** Reaction rate of redox mediators with singlet oxygen. Change in absorbance over time normalized to the initial value upon exposure to $^1O_2$ for DMPZ (**a**) and TTF (**b**) as extracted from the spectra in Fig. 1 at the respective peak maxima

B3LYP/6–31G* level. Vibrational frequencies were calculated to ensure that the optimized geometries were local minima and were used to determine the zero-point energies. The energies were adjusted for the well-known error in the $^1O_2$ at the DFT level compared to triplet $O_2$. In the case of the B3LYP/6–31+G* energies, this correction is 0.98 eV. The solution phase effects were included using a PCM continuum model[59] with a dielectric constant of 7.2. The reactions considered, the optimized structures of the products, and the reaction energies are shown in Fig. 4.

Considering first DMPZ, $^1O_2$ was reported to react with a diene via a Diels-Alder type [4 + 2] cycloaddition to yield endoperoxides. The electrophilicity of $^1O_2$ increases the reactivity of $^1O_2$ in the [4 + 2] cycloaddition towards aromatic compounds having a high electron density[50]. Thus, in general the RM will react more easily with $^1O_2$ than the RM+, as seen in the experimental data in Figs. 1–3. [4 + 2] cycloaddition at H-substituted aromatic carbons is much slower than at carbons with electron-donating groups such as $C_6H_5$, $CH_3$, or $OCH_3$, and thus, cycloaddition to 9,10-disubstituted anthracenes is favored over the other rings, and the reactivity follows the order $C_6H_5 < CH_3 < OCH_3$[50]. The subsequent reactions have been reported to be either cycloreversion or O–O bond cleavage, with the latter being preferred in the absence of substituents[50]. Translating this to DMPZ, the electron donation by the two amines may allow for appreciable reactivity at the 1,4 position for [4 + 2] cycloaddition to give the endoperoxide **5** (Fig. 4). Another possible route of attack is at the π-bond adjacent to the amine to yield the

dioxetane-type product **6**. However, we could not locate the corresponding biradical products that would form upon O–O cleavage in **5** as suggested for the decomposition of an endoperoxide[50]. [4 + 2] cycloaddition at DMPZ+ may also give the endoperoxide **7**. However, all these reactions are energetically rather unfavorable with large reaction barriers well above 1 eV. Oxidation to DMPZ+ appears to reduce the electron density at the 1,4 positions in a manner resulting in inferior reaction thermodynamics, which is also reflected in the even higher reaction barrier for DMPZ compared with DMPZ+ and seen in the experimentally observed faster reactivity of the former compared to the latter.

The reaction of $^1O_2$ with enes is the subject of longstanding mechanistic investigations and was found to be a complicated process with various possible intermediates such as a diradical, zwitterion, perepoxide, or dioxetane[48,49,51]. Typically, reactions of $^1O_2$ with substrates that contain an H at the α-C, e.g., $CH_3$–CH=$CH_2$, have been investigated, and found to lead to a shift of the double bond to form the hydroperoxide $CH_2$=CH–$CH_2$–OOH by H-abstraction at the methyl group[48,49]. The π-bonds of TTF, however, are all terminated by S, which may result in the reactive intermediates attacking further molecules. The ene position in the TTF rings is one possible position for $^1O_2$ attack, for which we found three possible head-on products, **8–10**, and the dioxetane **11**. While an analogous biradical could not be located for DMPZ, we were able to find the biradical **9** for TTF. The other possible point of attack is at the central C=C bond to form the dioxetane product **12**. Both reactions that lead to the dioxetanes are thermodynamically favorable, as they are exothermic and have rather small activation barriers of ~0.4–0.7 eV. Dioxetanes are known to undergo decomposition to the related carbonyl compounds by cleaving the C–C bond[60]. Reported mechanisms involve either [2 + 2] cycloelimination or a radical mechanism[61]. The sensitized photooxygenation of a related compound has been reported to undergo such cleavage to form the corresponding dithiocarbonate. The corresponding mechanism with TTF is shown in Supplementary Fig. 4 to form 1,3-dithiol-2-one. It is clearly seen in the NMR as the major newly formed decomposition product with a peak at 7.1 ppm. However, all the NMR visible products together after photooxygenation only equate to 40% of the initial TTF of which 28% are 1,3-dithiol-2-one. Hence, 1,3-dithiol-2-one decomposes further with $^1O_2$. There are minor peaks at ~5.8 and 1.2 ppm which we could not identify, but which are by far not accounting for all lost TTF. Alternative pathways where the organic sulfides react to give the corresponding peroxysulfoxides, sulfoxides, or sulfones **13–16**[55–57] appear less likely considering the high activation energies. Reactions with TTF+ could yield the product **17** through attack at the ring position or **18** through attack at the central position, with the latter being somewhat more favorable. For both mediators, the reactions of $^1O_2$ with the mediator cation are less favorable, because we find that $^1O_2$ forms molecular complexes with the cations that are more stable than the respective products of $^1O_2$ insertion.

Overall, the reaction energies in Figs. 4, 5 indicate that the TTF species have smaller barriers to reaction with $^1O_2$ than the DMPZ species, and thus will undergo faster reactions. This is consistent with the experimental studies of the reaction rates (Figs. 2, 3). We also find that the oxidized TTF has a higher reaction barrier than neutral TTF. This is consistent with the experimental finding that neutral TTF undergoes reaction with $^1O_2$ more readily than the oxidized state of TTF. Little difference was found between the reaction barriers of neutral DMPZ and DMPZ+, although we were not able to locate a structure for the biradical of DMPZ, which could be lower in energy than the closed shell singlet.

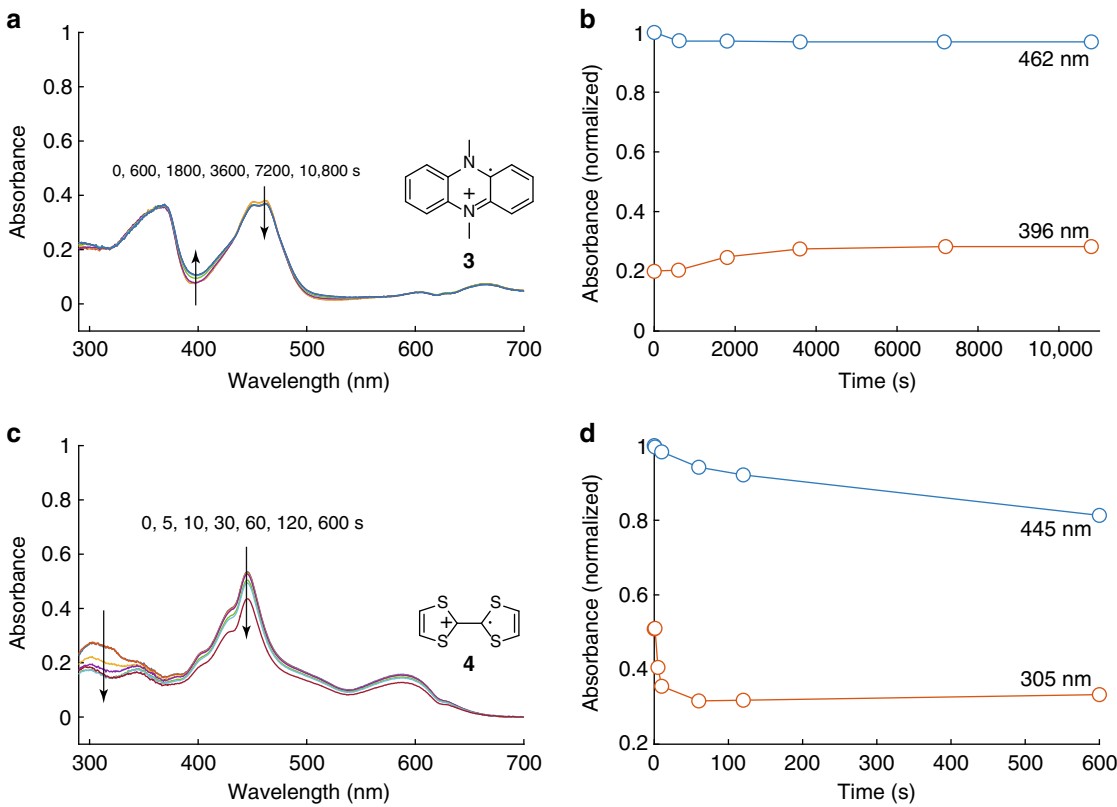

**Fig. 3** Stability of oxidized redox mediators against singlet oxygen. UV-Vis spectra of DMPZ$^+$ (**a**) and TTF$^+$ (**c**) upon exposure to $^1O_2$ as well as the normalized absorbance of DMPZ$^+$ (**b**) and TTF$^+$ (**d**) vs. time, as extracted from the positions indicated. DMPZ$^+$ or TTF$^+$ were generated by electrochemically oxidizing 0.02 M DMPZ or TTF, respectively in TEGDME electrolytes containing 0.1 M LiTFSI, followed by extraction into TEGDME to form a 250 µM solution. $^1O_2$ was photogenerated in the $O_2$-saturated solution using 1 µM palladium(II) *meso*-tetra(4-fluorophenyl)-tetrabenzoporphyrin and illumination at 643 nm, and the spectra were measured after the illumination times indicated. The spectrum of the sensitizer has been subtracted from **a** and **c**

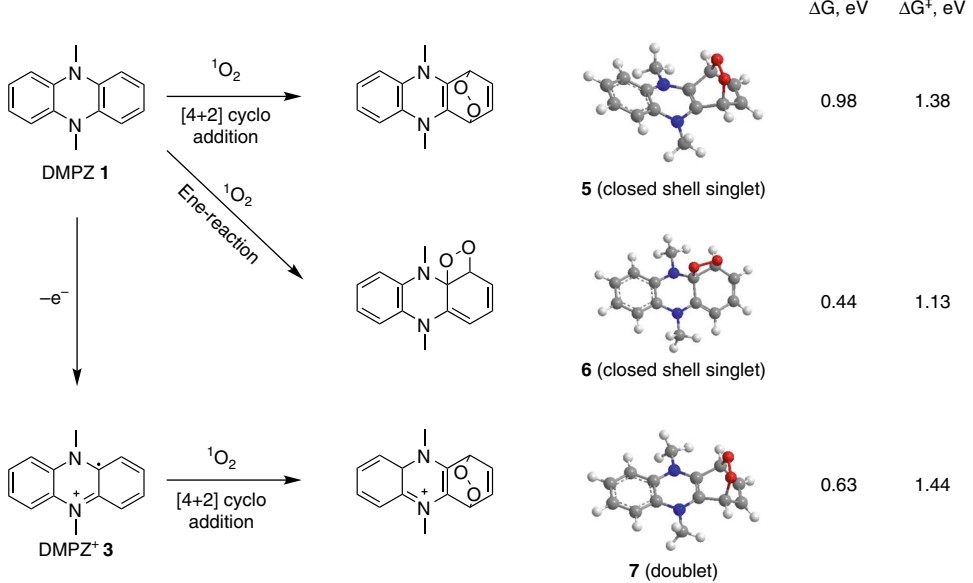

**Fig. 4** Reactions and energetics of reactions between DMPZ and $^1O_2$. Possible reactions of DMPZ/DMPZ$^+$ with $^1O_2$, B3LYP/6–31G* optimized geometries of the reaction products, and DFT-calculated free energies at 298 K (ΔG) and the barriers for these reactions (ΔG$^‡$)

## Discussion

The results of this study have multiple implications for required research directions in Li-O$_2$ cells towards developing practical energy storage devices. First, generally, the previous paradigm that stability of cell components against $O_2^-$ and $Li_2O_2$ were of prime importance needs to shift towards additionally and even more importantly stability against $^1O_2$. This concerns both studies on how materials degrade and on making more stable materials. Second, redox mediation is now widely accepted to be key for Li-O$_2$ batteries to achieve maximum energy density and

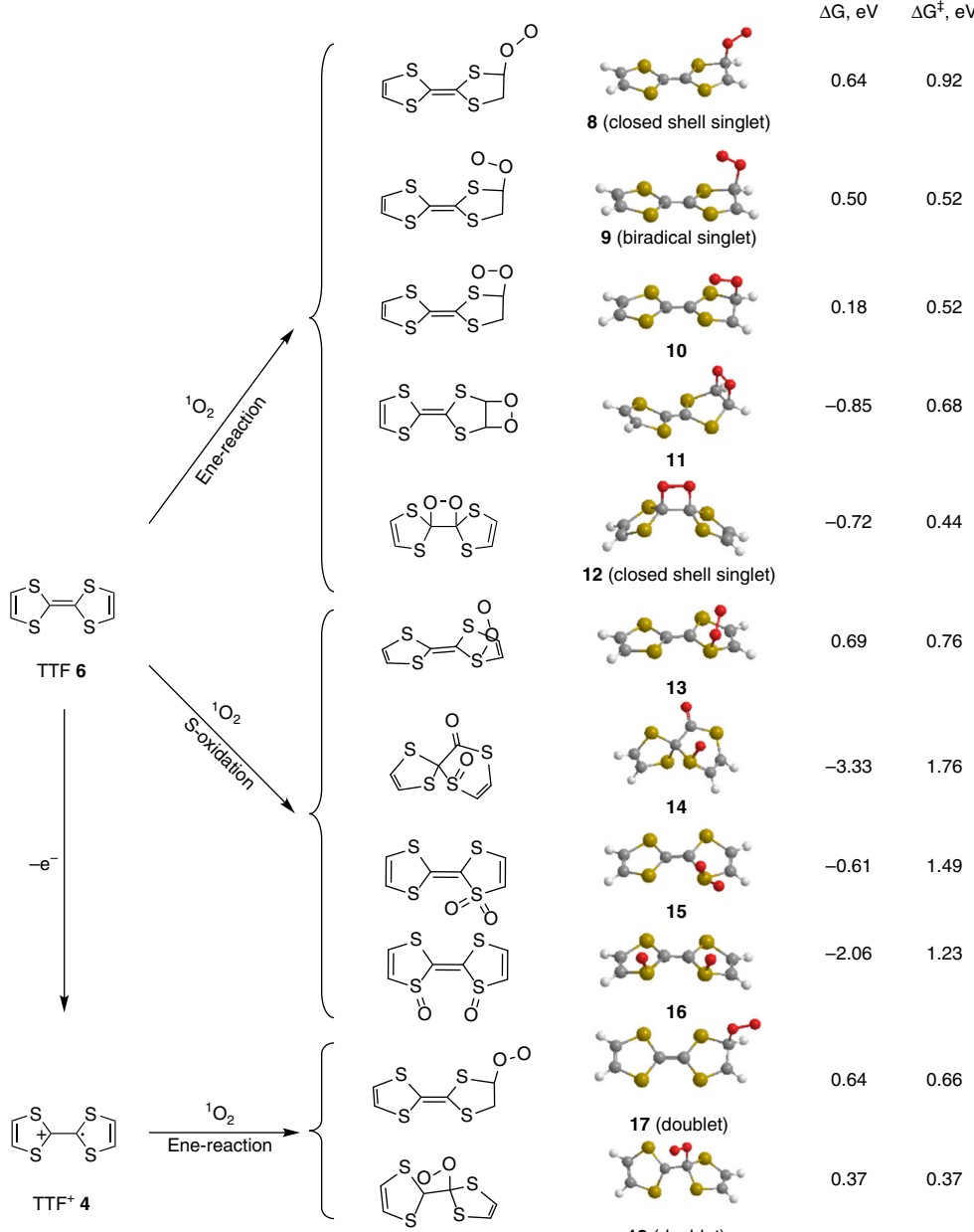

**Fig. 5** Reactions and energetics of reactions between TTF and $^1O_2$. Possible reactions of TTF/TTF$^+$ with $^1O_2$, B3LYP/6–31G* optimized geometries of the reaction products, and DFT-calculated free energies at 298 K ($\Delta G$) and the barriers for these reactions ($\Delta G^\ddagger$)

efficiency by far higher rates than possible without the mediator. The fact that we have shown that mediators can have very different susceptibility to decompose with $^1O_2$ spurs hope that even more stable mediators will be found. Third, the computational results that nicely reproduce the trend in reactivity between the investigated mediators and between the reduced and oxidized states suggest that computational screening will be a very effective tool to preselect candidate mediators. Fourth, even the best now available mediators may not be sufficiently stable for long term operation in presence of $^1O_2$. Therefore, additional means of counteracting degradation will likely be required. These may be chemical traps that more rapidly react with $^1O_2$ than other cell components, or, preferably, physical quenchers which catalyze the decay from $^1O_2$ to triplet oxygen[43].

In conclusion, we demonstrated that the widely observed gradual deactivation of RMs in Li-O$_2$ cells is predominantly caused by the decomposition of the RM by $^1O_2$. Thus, $^1O_2$-induced decomposition is the main reason for the decreasing catalytic effect of organic RMs during the cycling of Li-O$_2$ batteries, even when they are protected from the lithium metal. The reduced forms of the RMs are particularly vulnerable to $^1O_2$ attack because of the electrophilic nature of $^1O_2$. At the same time, we have shown that there are vast differences between the reactivities of different organic RMs, which spurs hope that RMs can be designed to be sufficiently stable for long-term operation. Therefore, the stability of RMs against $^1O_2$ must be considered, and RMs that are stable against $^1O_2$ attack must be found. Additionally, other measures to suppress $^1O_2$ formation by, e.g., quenchers are warranted.

## Methods

**Chemicals**. Tetraethylene glycol dimethyl ether (TEGDME, 99%), dimethoxy ethane (DME, 99%), bis(trifluoromethane)sulfonimide lithium salt (LiTFSI, 99.95%), dimethylphenazine (DMPZ), tetrathiafulvalene (TTF), potassium

superoxide ($KO_2$), lithium peroxide ($Li_2O_2$, 90%), 1,4,7,10,13,16-hexaoxacyclooctadecane (18-crown-6, ≥99%), acetonitrile (anhydrous, 99.8%) were purchased from Sigma-Aldrich. The lithium salt was dried in a vacuum oven for 3 days at 140 °C. The solvents were purified by distillation and further dried over activated molecular sieves. Palladium(II) *meso*-tetra(4-fluorophenyl)tetrabenzoporphyrin ($Pd_4F$) was synthesized as a sensitizer for $^1O_2$ generation according to a previously reported procedure[46]. Lithium iron phosphate ($LiFePO_4$) was purchased from MTI Corporation and used to prepare partially delithiated $LiFePO_4$ ($Li_{1−x}FePO_4$) according to a previously reported procedure[62].

**Electrochemical methods**. Electrolytes containing the oxidized RMs ($RM^+$) were prepared by electrochemical oxidation of the RMs. The electrochemical cells used to oxidize the RMs were based on a Swagelok design (see Supplementary Fig. 7). A porous carbon paper cathode (Freudenberg H2315), glass fiber separator (Whatman GF/F), and $Li_{1−x}FePO_4$ counter electrode were used in the cells, which contained 50 µL electrolyte (0.02 M RM (DMPZ or TTF) and 0.1 M LiTFSI in TEGDME). The working electrodes and separators were washed and dried at 120 °C for 24 h under vacuum prior to use. The $Li_{1−x}FePO_4$ counter electrodes were made by mixing partially delithiated active material with Super P and PTFE in the ratio 8:1:1(m/m/m) from which free-standing electrodes were obtained. The electrodes were vacuum dried at 200 °C for 24 h. The cells were assembled and operated using a MPG-2 potentiostat/galvanostat (BioLogic) in an Ar-filled glovebox. The cell containing DMPZ was charged at 100 µA to a potential of 3.5 V vs. Li/Li$^+$. The cell containing TTF was charged at 100 µA to 3.7 V vs. Li/Li$^+$.

**UV–Vis and $^1$H-NMR analysis**. UV–Vis absorption spectra were recorded on a Cary 50 UV–Vis spectrophotometer (Varian). For stability measurements against $O_2$, $KO_2$, $Li_2O_2$, and $^1O_2$ TEGDME electrolytes containing 0.1 M LiTFSI and 60 µM RM were used. For measuring $O_2$ stability, 2 mL of the electrolyte were saturated with a stream of pure $O_2$ via a septum for 10 min and then the solution further stirred for 24 h in a closed 20 mL vial with pure $O_2$ headspace. For measuring stability against $O_2^−$, 14.2 mg $KO_2$ and 52.9 mg 18-crown-6 (excess amount) were stirred in 2 mL of the electrolyte. For measuring stability against $Li_2O_2$, 1 mg $Li_2O_2$ were stirred in 2 mL of the electrolyte. The photochemical generation of $^1O_2$ was achieved by in situ photogeneration with the sensitizer $Pd_4F$. A $^3O_2$-saturated TEGDME electrolyte containing 0.1 M LiTFSI and 60 µM RM that contained 1 µM of the sensitizer was irradiated with a red light-emitting diode light source (OSRAM, 643 nm, 7 W). During the measurement, the electrolytes were stirred to ensure uniform RM and oxygen species concentration using a small size magnetic bar in the cuvette. The sample preparation for the oxidized RMs was carried out using the same procedure as for the RMs with the additional pre-oxidation process described above. After oxidation, the cells were disassembled immediately in the glovebox and the RM$^+$-containing electrolytes were extracted with TEGDME to obtain a total of 4 mL solution. The extract had thus a concentration of 250 µM RM$^+$, but likely somewhat less since the extraction from the porous media will be slow. One micrometer sensitizer was added for the following experiments with $^1O_2$. For the $^1$H-NMR measurements, samples were prepared as before but with a concentration of 1 mg RM in 1 mL DME. DME was used to allow for solvent evaporation. After the contact time with the oxygen species ($O_2$, $KO_2$, $Li_2O_2$, and $^1O_2$) the solvent was evaporated at room temperature under vacuum, the residue dissolved in 0.8 mL DMSO-$d_6$ and subjected to $^1$H-NMR measurement on a Bruker AVANCE III 300 MHz spectrometer. The DMSO peak is taken as internal reference for quantitative comparison of spectra.

## Data availability
The data that support the findings of this study are available from the corresponding author upon reasonable request.

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

## Acknowledgements

This work was supported by a Human Resources Development program (No. 20184010201720) of a Korea Institute of Energy Technology Evaluation and Planning (KETEP) grant, funded by the Ministry of Trade, Industry and Energy of the Korean government, and supported by National Research Foundation of Korea (NRF) grant funded by the Korea government Ministry of Education and Science Technology (MEST) (NRF-2018R1A2B3008794). S.A.F. is indebted to the European Research Council (ERC) under the European Union's Horizon 2020 research and innovation program (grant agreement no. 636069) and the Austrian Federal Ministry of Science, Research and Economy and the Austrian Research Promotion Agency (grant No. 845364) and initial funding from the Austrian Science Fund (FWF, Project No. P26870-N19). The work by L.A.C. and P.C.R. was supported by the U.S. Department of Energy, Office of Energy Efficiency and Renewable Energy, Vehicle Technologies Office.

## Author contributions

W.-J.K. conceived the concept and designed the experiments. W.-J.K., H.K., Y.K.P., C.L., T.T.N., N.M., H.-G.J. and S.M.B. performed the experiments. P.R. and L.A.C. performed the density functional theory calculations. S.A.F., W.-J.K., H.K., Y.K.P. and C.L. interpreted the data and wrote the manuscript. Y.-K.S. led the co-work for this study. Y.-K.S. and S.A.F. supervised this work.

## Additional information

**Competing interests:** The authors declare no competing interests.

