## [Peer Review File · Nature Communications]

Reviewer #1 (Remarks to the Author):

Please see attached document (Please be noted that there is a typo in this report. 'Nature Chemistry' should be 'Nature Communications')

(Review) Nature Chemistry: Deactivation of redox mediators in Li-O₂ batteries by singlet oxygen

Summary

The paper aims to demonstrate that the redox mediators (DMPZ and TTF) undergo degradation reactions in the presence of singlet oxygen but are reasonably stable in the presence of oxygen, potassium superoxide, and lithium peroxide.

Much of the experimental focus employs a palladium based photosensitizer to generate in situ singlet oxygen. UV-Vis is used to demonstrate that the mediators degrade in the presence of the formation of singlet oxygen. NMR spectra are included as supplementary information. The paper also claims to form DMPZ/TTF radical cations by electrochemical means and then examine their respective chemistries. The second part of the paper focuses on computational efforts to examine the degradation reaction of DMPZ and TTF (and their respective radical cations) by singlet oxygen.

What follows is a summary of some major problems with the manuscript-

- A) Rate constants are calculated from exponential fits of the UV-vis spectra that do not appear to exhibit exponential (1st order) kinetics. This particular aspect of the paper should be removed. Alternatively, the authors should be sure to give a rigorous justification of the appropriateness of the kind of analysis. They should also explain how the data were fitted.
- B) Despite assertions to the contrary, the NMR spectra for DMPZ reported in the supplementary information (Figure 3) does not appear consistent with the UV-Vis spectra reported in the main text. Furthermore, based on the spectra of the "blank" DMPZ presented, I have serious reservation about the purity of the starting material DMPZ. The presented spectrum does not appear to be "clean" and looks very similar to what is asserted to be 20 % degraded DMPZ. (see below for further explanation)
- C) Also in the supplementary information (Figure 2), there appears to be absolutely no DMPZ in the proton NMR spectra. The spectrum also looks completely different from the spectra given in Figure 3 of the supplementary information.
- D) I am less qualified to evaluate the computational work since this is outside my expertise. However, several of the ChemDraw structures do not appear to correspond to the computational structures shown in Figure 5. Second there is a weak connection between the computational work and the experimental work. The experimental work indicates that singlet oxygen degrades these aromatic redox mediators. As cited in the manuscript, previous literature indicates a reactions do occur between aromatic compounds and singlet oxygen. The computational work describes formation of sulfones, sulfoxide, and organic peroxide degradation products. The computation work

also tries to rationalizes the reason for the greater ability of neutral TTF to react with singlet oxygen compared to the corresponding TTF radical cation. Unfortunately, the experimental work makes little attempt to demonstrate the formation of these degradation products. Isolation and chemical characterization of one or two of the degradation products will allow you to focus your computational efforts. Without this, the computational work appears to be disconnected from the rest of the manuscript for this reason.

- E) Finally, the experimental section and details given throughout the manuscript is really quite poor. These experiments do not seem particularly complicated to carry out. However, key details like solvents used and concentration of different species are scattered throughout, absent altogether, or contradictory. As an example, three different wavelengths were given for the irradiation of the photosensitizer. I do not believe that I could reliably replicate the described experiments.

These major points are recapitulated in greater detail in the comments below.

I cannot recommend publication in the journal. The experimental work is not of sufficient quality (see above and below). The computational work is somewhat disconnected from experimental evidence- many structures/degradation products are calculated but none of these have been shown to actually be present or relevant in the experimental work.

Results Section

- A) Lines 81-83 the statement “DMPZ has one of the lowest charge potentials reported and has been shown to be thermodynamically stable based on DFT calculation and experimental verification. 23”

This statement is not clear-

The referenced paper claims (as shown by DFT and experimental work) that the oxidation potential of DMPZ is higher than that of lithium peroxide but lower than that of the electrolyte and therefore is capable of fulfilling its role as a redox mediator while not facilitating the decomposition of the electrolyte.

Has DMPZ been shown to be thermodynamically (or kinetically) stable to the presence of lithium metal? The statement as written in the text is somewhat vague.

- B) Line 93, illumination with what? What wavelength? The first mention of this detail is in the caption (at 695 nm wavelength) but you should mention this wavelength in the text on line 93 (and the experimental section). Later in the experimental section you give another wavelength and in the supplementary information a 3rd wavelength is given!

C) Line 94- the compound has “meso” italicized but this same compound is not italicized on Lines 260 Pick a consistent standard

D) Line 102, “The NMR intensity of DMPZ decreased by ~20 %, in agreement with the similar decrease observed in its UV absorbance (Figure 1z and Supplementary Fig 3b)”

Your NMR spectra as presented in your supplementary information does not support the statement in the main text. First, the spectra nearly overlap, *prima facie* there doesn't appear to a 20 % difference in area. Second, there does not appear to be an internal standard or reference peak that would allow for a quantitative assessment of the concentration of DMPZ as measured by proton NMR. Details of quantification by NMR are not given in the experimental. Thirdly, assuming that there is 20% degradation and that these degradation products are structurally similar then there should be *new* peaks in the aromatic region that are not present in the spectrum labeled “blank”. It seems that your “3h” spectra and your “blank” spectra are not distinguishable.

Finally, the authors should determine if their sample of DMPZ is actually pure. There should be two aromatic resonance (multiplets). The spectra labeled “blank” in Supplementary Fig 3B indicates at least three resonances (at 6.6 ppm, 6.8 ppm, and 7.1 ppm). Are you certain that your starting material is actually pure or is it already degraded as purchased? I should note that this doesn't appear to be a problem with you TTF sample shown in Figure 2b.

Along these lines the NMR is cut-off at 4 ppm. Is this due to the presence of TEGDME? The entire spectra should be shown. Where methyl resonances at ~3 ppm for DPMZ? Why are the spectra cut at 4 ppm? This region is particularly important since one might expect these functional groups to be oxidized!

I would also be sure to take a reference spectrum of DMPZ to ensure that the starting material is sufficiently pure.

E) Fig 1- The “O2, KO2, and 1O2” labels in Fig a,b,c should be copied in Fig d,e,f as this would make it easier to comprehend instead of having the reader trying to figure this out on his own

F) Fig 2. In Figure 1c you listed a convenient measurement time in minutes? and a measurement time of seconds for Figure 1f. Perhaps consider the same units of time for all graphs?

Also where spectra taken continuously at 342, 450, 360, and 308 nm? The graphs would seem to imply that is the case. However your SI figure S4 (and Figure 1c,f) indicates that the spectra were taken at discrete points. Your graphs in Figure 2 should reflect this with markers indicating individual data points (along with an interpolating line if you want).

G) Table 1- the description has some problem that do not render correctly- be aware this may be a problem for publication.

H) "Table 1. Kinetics of the decomposition of DMPZ and TTF and their cation with 1O₂.

Values are given as the slope at [gibberish] in s⁻¹ and are extracted from the exponential fits [gibberish] given in Fig. S3. "

Similar problems occurs on line 170—"Thus, 1O₂ produces R-OOH, R[gibberish], and R-OO[gibberish]..."

I) Comments on Figure 2 and Table 1..

The authors attempt to extract kinetic constants from their UV-Vis spectra by making exponential fits. According to Beer's law absorbance (the logarithm of transmission) is proportional to the concentration of that species. The plots in figure 2 at 342 nm, 450 nm, and 360 nm do not appear to follow exponential growth or decay during the time of the experiment. (The TTF graph at 308 nm is an exception). It is not appropriate to extract and report rate constants from exponential rate constant. The entire table with rate constants should be discarded. Is sufficient to explain that TTF degrades much more readily than DMPZ as evidenced in Fig 1.

J) In line 141, you state "we oxidized the mediators electrochemically", How exactly was this done-electrodes used? Concentration? Electrolyte solution, etc? What is the counterion to the radical cation (DMPZ⁺, TTF⁺) . Furthermore there is no indication as to the concentration of these radical cations in solution.

K) Line 150-151 " the DMPZ/DMPZ⁺ redox couple is degraded by 1O₂, because DMPZ reaction strongly, which degrades the DMPZ/DMPZ⁺ couple as a whole" This is tautological sentence. The same thing is said twice by saying the same thing in two different ways.

L) The structure for DMPZ⁺ is presumably a radical cation such as shown for TTF⁺ and is drawn incorrectly in Fig3a

M) Again are the data for Fig 3b, and Fig 3d continuous or discrete? Marker points should be used to indicate specific data points—there should not be just a simple line

N) Line 169 should "peroxides" be replaced with "an organic peroxide" or even "peroxide"

O) Line 173, there is an extra space "and TTF/TTF⁺ , respectively,"

P) Lines 175, "We also carried out density functional (DFT)..." should be written as "We also carried out density functional theory (DFT)..."

- Q) Line 190 “(Ref. 50)” should be “⁵⁰”
- R) Fig 4- Species 4 is drawn is not drawn as a radical cation. Also it is a labeled species but was not labeled earlier in Fig 3a
- S) Fig 5, Species 13 appears to be a rearranged product where one of the 5 membered rings has shifted to a 6 membered ring. To form some kind of sulfoxide and ketone(?). There is no/little discussion in the text
- T) Fig 5, There should be a ChemDraw for each molecule presented—it is confusing when you have only one diagram that supposedly represents multiple species (7, 8, 9, and 10) (and 12, 13, 14, an 15)
- U) Line 224, “The possibility for these products is supported by the NMR results in supplementary Fig 3b, which can be correlated to the formation of such products at one or multiple S-atoms. “

This is a rather weak statement, all that is shown in Figure 3b is that the TTF has decomposed or degraded as a consequence of the reaction conditions. The NMR does not give indication as to the nature or identify of the products. If anything, the fact that the spectrum is so simple indicates that your degradation products may be represented by 2 or 3 major compounds and that a reasonable chance of isolation of the degradation products exists.

I would suggest the use of ESI-mass spectroscopy on the crude mixture to obtain the molecular weight of the unidentified products. Gas chromatography (with a mass spectrometer) would likely further degrade these presumed organic peroxides. The employment preparative (silica gel) TLC likely isolate and fully characterize these degradation products. This would allow you to focus your computation efforts on specific identified products.

- V) Line 257 missing comma “(18-crown-6 >99%)”

Electrochemical methods

- A) Line 266-269 “ The electrochemical cells used to oxidize the RMs were based on a Swagelok design....”

This experimental description is vague. I have no idea what “Swagelok design” means other than you bought parts from a company—is there a better reference which explains your cell design. Also what kind of carbon cathode was use? What kind of glass fiber separator was used? What are the dimensions? The capacities? How much electrolyte was used?

- B) Lined 275- "After the oxidation of the RM, the cells were disassembled immediately in the glovebox... extracted for the following experiments..." Into what were they extracted into? What solvent? At what concentration?
- C) Furthermore, you mention the purification of LiTFSI but nowhere in the manuscript or experimental do you mention its use or its concentration. Presumably it was used as the electrolyte species in a solvent but at what concentration?
- D) Was TEGDME used as a solvent? There isn't any mention of its use in your experimental writeup.
- E) Was acetonitrile ever used as a solvent? There isn't any mention of its use in your experimental writeup or anywhere in the paper for that matter.
- F) Line 269 "(DMPZ and TTF)" should be "(DMPZ or TTF)"

UV Vis and NMR analysis

- A) Line 281 "were added to 2 mL of the RM-containing electrolytes and diluted before measurement using purified solvents".
What solvents were used acetonitrile or TEGDME? What was the concentration of the RM mediator? Was LiTFSI used in the electrolyte? If so at what concentration and at what concentration?
- B) What was the concentration of the sensitizer used?
- C) What is OSRAM? Be nice to know if this is the company for the diode and what is the model number?
- D) How were the lithium peroxide experiments carried out? There are indications in the main text of the paper but explicit details should be given in this section.
- E) How were the oxygen experiments carried out? Was a solution made and allowed to sit on the benchtop? Was oxygen bubbled throughout the solution? Was the solution put in a vessel with an O₂ atmosphere, sealed and stirred for 24 h?

Reference section

- A) Line 297 - Reference 1 has a missing author.

- B) On Line 317- only one page number is listed “1400867”; is this the correct format? The same can be said for Line 346, 351 370,377, 353, and 405 most references are cited with the range of page number- for example “J. Phys. Chem. Lett. , 2989-2993.”
- C) Most of the references give the full authorship list. Many of the other references give a single author- for example “Mahne, et al” Is this the correct format?
- D) On Line 340, there is no space after the comma “IMLB,Jeju, Korea, S6-3 (2012).”
- E) On Line 384, there is a typo “L-Air Batteries”
- F) On Line 399, the full authors names are given “Last Name, First Name” instead of the normal abbreviation “Last Name, FN”
- G) On Line 428, The “n” for “n-heptane” should be in italics?
- H) On Line 437, there is an extra space “Density functional”

Supporting Information

- A) Figure 1a --- “DMPZ in G4 baseline MeCN”

What does this mean? What is G4- do you mean TEGDME? Why is MeCN mentioned? Your caption indicates TEGDME is the solvent. You should either use G4 or TEGDME not both.

The method by which you expose your mediators to lithium peroxide is described in your caption and to some degree in your text but is absent in your experimental section.

Presumably when you present your data the spectra indicating exposure to lithium peroxide after 24h is offset by 0.3 absorbance units. If this is the case, then the number scale should be deleted since your absorbance values for the offset spectra are arbitrary. Your caption should also make mention of this offset.

- B) Figure 2

What kind of NMR spectra is this? It should say “¹H-NMR “ in the caption. Your caption gives experimental details that appear at variance to your experimental details given in your experimental section. Was the procedure different for Uv-Vis KO₂/Li₂O₂ experiments compared to NMR experiments KO₂/Li₂O₂.

What was the NMR solvent? Is it CDCl₃/TEGDME or CDCl₃ or TEGDME?

You are missing quite a few peaks in your spectra---

If you used CDCl₃ as your NMR solvent there should be a CHCl₃ peak around 7.26 ppm. It seems that the spectra have been purposely cut at ~7.2 ppm.

Where are the aromatic resonances attributable to DMPZ at around ~7 ppm? Where are the methyl resonances (~3 ppm) attributable to DMPZ? Have they been cut off? Looking at Figure 2a there doesn't appear to be any DMPZ in your spectrum.

If you used TEGDME as indicated in the caption where are the resonances attributable to the ether?

What is this broad resonance around 6.5 to 6.0 ppm? Is it protonated TFSI salt?

In Figure 2b, where are the peaks attributable to CHCl₃ or TEGDME? You should show the full spectra? If you wish to show only the spectrum from 4.0 ppm to 7.2 ppm you can do that with an inset or a second or third figure

C) Figure 3

Again there should be an indication that this is a proton spectra. Your experimental section indicates that CDCl₃ was used but there is no indication in the caption.

In the caption is the first mention of the concentration of the sensitizer. (1 μM).

I would consider putting a reference spectra for the sensitizer since it would assist the reader in peak identification (or lack of peaks).

So at what wavelength was the sample irradiated at? The text mentions 695 nm. Your experimental section mentions 643 nm. The caption in your supplementary indicates 625nm.

Figure 3a There appears to be negligible change in the spectra. This result is very different that your UV-Vis experiments would indicate. In Figure 2 of your main text, you assigned 342 nm as an absorbance for DMPZ. Over the course of 3 h the absorbance value for this wavelength dropped from 1.0 to 0.8. This would indicate a change of 20 % in DMPZ concentration yet there doesn't appear to a corresponding change in the NMR spectra. Admittedly there is no internal standard for your NMR spectra, however there doesn't even appear to the growth of any new peaks in your NMR spectra consistent with related degradation products. Where is the missing 20 %? This discrepancy should be explained somewhere in your manuscript

Reviewer #2 (Remarks to the Author):

General impression

Freunberger and Sun et al. reported on the deactivation of redox mediator (RM) for lithium-oxygen cells caused by singlet oxygen, via a combined experimental and computational approach. This cutting-edge research is of utmost importance to the field of metal-oxygen batteries, as it identifies the culprit of RM degradation, which is a long-standing puzzle, and therefore points out a promising pathway to diminish or radically eliminate RM degradation. The manuscript is clearly written. The major arguments are well grounded on experimental facts. Considering these points, this reviewer is happy to recommend publication of this work after minor revisions that I believe will further improve the quality of this work.

Two suggestions

(1) Further discussion of the implications and their demonstration in a lithium-oxygen cell

At its present state, this work mainly compares the reactivity of O₂, superoxide and singlet oxygen towards two RMs and then dwells upon the reaction mechanism. Discussion on the implication of this study to lithium-oxygen cells is refrained. An important missing part to collaborate the conclusion that singlet oxygen is the major culprit of RM degradation is to demonstrate that by somehow trapping singlet oxygen the lithium-oxygen battery with RM has a longer lifetime.

(2) Reorganization of the results section and extension of the discussion section

The presented manuscript has a very short discussion section. I found that extensive discussion is actually contained in the results section. I would suggest the authors to move those parts to the discussion section.

Response to reviewers on the paper entitled

“Deactivation of redox mediators in Li-O₂ batteries by singlet oxygen”

manuscript: NCOMMS-18-24458

We thank the Reviewers for their helpful comments to which we respond below point by point. The Reviewers' comments are reproduced in italics. In the manuscript and supporting information we highlighted the changes in yellow. The comments helped us to greatly improve the manuscript and we are confident that we have satisfyingly responded to all of them.

Reviewer #1:

Summary

The paper aims to demonstrate that the redox mediators (DMPZ and TTF) undergo degradation reactions in the presence of singlet oxygen but are reasonably stable in the presence of oxygen, potassium superoxide, and lithium peroxide. Much of the experimental focus employs a palladium based photosensitizer to generate in situ singlet oxygen. UV-Vis is used to demonstrate that the mediators degrade in the presence of the formation of singlet oxygen. NMR spectra are included as supplementary information. The paper also claims to form DMPZ/TTF radical cations by electrochemical means and then examine their respective chemistries. The second part of the paper focuses on computational efforts to examine the degradation reaction of DMPZ and TTF (and their respective radical cations) by singlet oxygen.

What follows is a summary of some major problems with the manuscript-

We thank the Reviewer for the very careful and insightful review, which greatly helped us to improve the manuscript and to make the messages clear. In response to your comments, we have added all the requested new data, discussion, and experimental details and revised both the manuscript and the supporting information accordingly. All these changes make us confident that the this way improved manuscript will make a convincing case that singlet oxygen is the dominant cause for redox mediator degradation in Li-O₂ cells.

A) Rate constants are calculated from exponential fits of the UV-vis spectra that do not appear to exhibit exponential (1st order) kinetics. This particular aspect of the paper should be removed. Alternatively, the authors should be sure to give a rigorous justification of the appropriateness of the kind of analysis. They should also explain how the data were fitted.

Thank you for this comment. First of all, we apologize that the used equations did not render properly in the manuscript you had at hand (as mentioned in comment H) on the “Results Section”) even though our downloaded pdf-files looked fine.

Please note that we did not imply any particular reaction order by the fits, but what we attempted was simply to get a means of comparing consumption rates of reduced and oxidized forms of the DMPZ and TTF. To do so, we considered that the slope of the absorbance versus time at the start as given in Fig. 2 would be appropriate. To obtain this slope from discrete experimental data, we figured that a fit of the data with an equation that *phenomenologically* fits the data would be appropriate and thus we used an equation of the form $A(t) = a \cdot \exp(-b \cdot t) + c$, with A being absorbance. Any other regression

analysis like, for example, cubic splines would have given a similar fit. The data in Table 1 are simply the slope of these fits at $t = 0$ s. They are not rate constants.

In response to your more detailed comments G) to I) in the “Results Section” we removed this table and state as you suggested simply that TTF decays much more rapidly than DMPZ and also that the reduced form of each mediator decays more rapidly than the oxidized form.

B) Despite assertions to the contrary, the NMR spectra for DMPZ reported in the supplementary information (Figure 3) does not appear consistent with the UV-Vis spectra reported in the main text. Furthermore, based on the spectra of the “blank” DMPZ presented, I have serious reservation about the purity of the starting material DMPZ. The presented spectrum does not appear to be “clean” and looks very similar to what is asserted to be 20 % degraded DMPZ. (see below for further explanation).

Thank you for the close inspection of the NMR part. In response to your insightful comments we completely redid all $^1\text{H-NMR}$ analysis. Previous spectra were obtained by using the same $60\ \mu\text{M}$ mediator solutions in $0.1\ \text{M LiTFSI}$ in TEGDME as used for UV-Vis analysis in the main paper and by dissolving them in CDCl_3 . The low concentration of mediator in combination with the far overwhelming amount of TEGDME and the CDCl_3 caused all the problems you mentioned. The previous spectra should hence be disregarded.

Consequently, we redid all $^1\text{H-NMR}$ analysis with the following changes: (i) We now use $1.25\ \text{mg}$ mediator per mL NMR solvent to have a much better signal-to-noise ratio. (ii) At this concentration it became apparent that CDCl_3 is not suitable since particularly the DMPZ spectra show extremely broad peaks; this was the reason why DMPZ in the previous Fig. S2a has shown only one extremely broad peak (the broad resonance around 6.5 to $6\ \text{ppm}$ you mentioned). We used DMSO-d_6 instead, which allows for well resolved peaks. (iii) Instead of TEGDME we used dimethoxyethane (DME), which we removed by evaporation and hence allows for the higher mediator concentration in the NMR sample. All details of the procedures are mentioned in the updated Methods section.

Regarding the purity of DMPZ and the degradation by $^1\text{O}_2$: the new $^1\text{H-NMR}$ spectra in the new Fig. S3a (Fig. R1 below) show that the DMPZ is pure; only peaks assigned to DMPZ and DMSO-d_6 with trace water are present. Please note that this is the identical DMPZ as used before without any further purification. Taking the DMSO peak as internal reference, the spectra of the samples exposed to O_2 , KO_2 and Li_2O_2 (new Fig. S2) show no noticeable change in concentration in accord with the UV-Vis results. Analogously, the spectrum of the sample after $3\ \text{h}$ exposure to $^1\text{O}_2$ (Fig. R1) shows a decrease of DMPZ concentration in accord with the UV-Vis spectra in the main text.

Figure R1. (new Fig. S3) Stability of DMPZ in contact with $^1\text{O}_2$. NMR spectra (in DMSO-d_6) were recorded before and after photooxygenation. 1 mg DMPZ in 1 mL in 0.1 M LiTFSI/DME were saturated with O_2 . For photochemical $^1\text{O}_2$ generation, 1 μM of the photosensitizer palladium(II) meso-tetra(4-fluorophenyl)-tetrabenzoporphyrin (Pd4F) was dissolved in the solution and illuminated with 643 nm radiation for 3h.

C) Also in the supplementary information (Figure 2), there appears to be absolutely no DMPZ in the proton NMR spectra. The spectrum also looks completely different from the spectra given in Figure 3 of the supplementary information..

The answer to this comment links to the one above. CDCl_3 turned out to be unsuitable for DMPZ since spectra show extremely broad peaks. For example the two peaks for the aromatic H that are now seen in Fig. R1 above had merged into the extremely broad peak that was visible between 6.5 and 6 ppm. We thus have redone all $^1\text{H-NMR}$ analysis as described above. More detailed responses are given with comments below.

D) I am less qualified to evaluate the computational work since this is outside my expertise. However, several of the ChemDraw structures do not appear to correspond to the computational structures shown in Figure 5. Second there is a weak connection between the computational work and the experimental work. The experimental work indicates that singlet oxygen degrades these aromatic redox mediators. As cited in the manuscript, previous literature indicates a reactions do occur between aromatic compounds and singlet oxygen. The computational work describes formation of sulfones, sulfoxide, and organic peroxide degradation products. The computation work also tries to rationalizes the reason for the greater ability of neutral TTF to react with singlet oxygen compared to the corresponding TTF radical cation. Unfortunately, the experimental work makes little attempt to demonstrate the formation of these degradation products. Isolation and chemical characterization of one or two of the degradation products will allow you to focus your computational efforts. Without this, the computational work appears to be disconnected from the rest of the manuscript for this reason.

We have corrected the ChemDraw structures that did not correspond to the computational structures. Please note that it is difficult to exactly represent the computed structures in ChemDraw because there are some bonds that are between single/double and electrons distribution are not all integer.

The point of the computational work was to calculate barriers for *onset* reactions to provide an understanding for the differences in stability for singlet O₂ reactions. The calculations have provided important mechanistic insight into the experimental studies and are consistent with the results. An experimental project to characterize the specific degradation products is outside the scope of this work and is not needed to achieve the goals of the work. The structures from the computations provide insight into the reaction mechanisms that are difficult to obtain experimentally. As common for oxidative decomposition reactions, rarely defined products are found¹. It is also typical for decomposition reactions that the onset step is typically the most demanding. The computed species represent likely the products of the onset reactions and will further decompose into a plethora of possible products. Therefore, even isolating a product, if at all possible, would not give a clearer indication of onset reactions.

The clear link between the computational and experimental work is the well reproduced trend of degradation rates of DMPZ vs. TTF and the reduced vs. the oxidized forms. This demonstrates that the found onset reactions are representative for the actually occurring ones.

5) Finally, the experimental section and details given throughout the manuscript is really quite poor. These experiments do not seem particularly complicated to carry out. However, key details like solvents used and concentration of different species are scattered throughout, absent altogether, or contradictory. As an example, three different wavelengths were given for the irradiation of the photosensitizer. I do not believe that I could reliably replicate the described experiments..

We apologize for not having given enough details in the experimental section. In response to your comments, we comprehensively improved the experimental section and give all details to reliably replicate the experiments.

These major points are recapitulated in greater detail in the comments below.

I cannot recommend publication in the journal. The experimental work is not of sufficient quality (see above and below). The computational work is somewhat disconnected from experimental evidence- many structures/degradation products are calculated but none of these have been shown to actually be present or relevant in the experimental work

We again thank you very much for the thorough review and insightful comments above and below. They greatly helped us to significantly improve the manuscript. Based on them, we are confident that we could clear out all concerns about the experimental work. In response to your comments we completely redid the ¹H-NMR analysis, which further supports the conclusions from UV-Vis about stability or instability of the mediators against O₂, superoxide, peroxide and ¹O₂. They further suggest that, as common for oxidative decomposition reactions, 1O₂ induced decomposition leads to a plethora of possible products. The fact that the found energetics of the onset reactions resembles the experimentally found relative decomposition rates established a clear link between the experimental and computational work.

With all this, we are confident that the significance of the work for the further development of Li-O₂ battery becomes now clear.

¹ See, e.g., Curran et al. *Combust. Flame* 114, 149, (1998).

Results Section

A) Lines 81-83 the statement “DMPZ has one of the lowest charge potentials reported and has been shown to be thermodynamically stable based on DFT calculation and experimental verification. 23” This statement is not clear-

The referenced paper claims (as shown by DFT and experimental work) that the oxidation potential of DMPZ is higher than that of lithium peroxide but lower than that of the electrolyte and therefore is capable of fulfilling its role as a redox mediator while not facilitating the decomposition of the electrolyte.

Has DMPZ been shown to be thermodynamically (or kinetically) stable to the presence of lithium metal? The statement as written in the text is somewhat vague.

We revised the text to make clear that the shown thermodynamic stability in the cited paper only refers to compatibility with the electrolyte.

B) Line 93, illumination with what? What wavelength? The first mention of this detail is in the caption (at 695 nm wavelength) but you should mention this wavelength in the text on line 93 (and the experimental section). Later in the experimental section you give another wavelength and in the supplementary information a 3rd wavelength is given!

We apologize for not having given enough detail at this point and about the confusion with the wavelengths (also your comment C) in “Supporting Information”). The correct wavelength is 643 nm. We clarified the text here and in all other places where illumination is mentioned.

C) Line 94- the compound has “meso” italicized but this same compound is not italicized on Lines 260 Pick a consistent standard

We made all occurrences of the word “meso” italic.

D) Line 102, “The NMR intensity of DMPZ decreased by ~20 %, in agreement with the similar decrease observed in its UV absorbance (Figure 1z and Supplementary Fig 3b)”

Your NMR spectra as presented in your supplementary information does not support the statement in the main text. First, the spectra nearly overlap, prima facie there doesn't appear to a 20 % difference in area. Second, there does not appear to be an internal standard or reference peak that would allow for a quantitative assessment of the concentration of DMPZ as measured by proton NMR. Details of quantification by NMR are not given in the experimental. Thirdly, assuming that there is 20% degradation and that these degradation products are structurally similar then there should be new peaks in the aromatic region that are not present in the spectrum labeled “blank”. It seems that your “3h” spectra and your “blank” spectra are not distinguishable.

Finally, the authors should determine if their sample of DMPZ is actually pure. There should be two aromatic resonance (multiplets). The spectra labeled “blank” in Supplementary Fig 3B indicates at least three resonances (at 6.6 ppm, 6.8 ppm, and 7.1 ppm). Are you certain that your starting material is actually pure or is it already degraded as purchased? I should note that this doesn't appear to be a problem with you TTF sample shown in Figure 2b.

Along these lines the NMR is cut-off at 4 ppm. Is this due to the presence of TEGDME? The entire spectra should be shown. Where methyl resonances at ~3 ppm for DPMZ? Why are the spectra cut at 4 ppm? This region is particularly important since one might expect these functional groups to

be oxidized! I would also be sure to take a reference spectrum of DMPZ to ensure that the starting material is sufficiently pure.

Please note the explanation given in comment B) of the “Summary” above. The poor quality of the previous spectra was due to unfavourable sample preparation. In response to your comments we redid all $^1\text{H-NMR}$ analysis with changed sample preparation and NMR solvent. The spectra for DMPZ degradation with $^1\text{O}_2$ (see Fig. R1 below) now support the conclusions from the UV-Vis measurements. We used the DMSO peak as internal standard to quantitatively compare spectra. The blank spectrum confirms that the DMPZ starting material was pure; only peaks assigned to DMPZ and DMSO- d_6 with trace water are present.

We also show now the entire spectra. The reason why we had previously cut the spectra to 4–7.2 ppm was that TEGDME and CDCl_3 peaks would by far have dominated the spectra.

Figure R1. (new Fig. S3) Stability of DMPZ in contact with $^1\text{O}_2$. $^1\text{H-NMR}$ spectra (in DMSO-d_6) were recorded before and after photooxygenation. 1 mg DMPZ in 1 mL in DME were saturated with O_2 . For photochemical $^1\text{O}_2$ generation, 1 μM of the photosensitizer palladium(II) meso-tetra(4-fluorophenyl)-tetrabenzoporphyrin (Pd4F) was dissolved in the solution and illuminated with 643 nm radiation for 3h.

E) Fig 1- The “O₂, KO₂, and 1O₂” labels in Fig a,b,c should be copied in Fig d,e,f as this would make it easier to comprehend instead of having the reader trying to figure this out on his own

We copied the labels “ O_2 , KO_2 , and $^1\text{O}_2$ ” also into Fig. d, e, f to make this point clear.

F) Fig 2. In Figure 1c you listed a convenient measurement time in minutes? and a measurement time of seconds for Figure 1f. Perhaps consider the same units of time for all graphs?

Also where spectra taken continuously at 342, 450, 360, and 308 nm? The graphs would seem to imply that is the case. However your SI figure S4 (and Figure 1c,f) indicates that the spectra were

taken at discrete points. Your graphs in Figure 2 should reflect this with markers indicating individual data points (along with an interpolating line if you want).

We now give all times in seconds to make them easier comparable. For the graphs in Fig. 2 we extracted the absorbance values from the spectra in Fig. 1. Values were thus not taken continuously but only at the discrete points indicated in Fig. 1. To make this clear, we added markers in Fig. 2 to make this clear.

G) *Table 1- the description has some problem that do not render correctly- be aware this may be a problem for publication.*

H) *“Table 1. Kinetics of the decomposition of DMPZ and TTF and their cation with IO₂. Values are given as the slope at [gibberish] in s⁻¹ and are extracted from the exponential fits [gibberish] given in Fig. S3. “Similar problems occurs on line 170—“Thus, IO₂ produces R-OOH, R[gibberish], and ROO[gibberish]...””.*

We apologize for the problems with rendering in the reviewer available file. The file available to us during submission to check correct display looked correct. Since we removed the table in response to the reviewer’s suggestion this problem is resolved.

I) *Comments on Figure 2 and Table 1. The authors attempt to extract kinetic constants from their UV-Vis spectra by making exponential fits. According to Beer’s law absorbance (the logarithm of transmission) is proportional to the concentration of that species. The plots in figure 2 at 342 nm, 450 nm, and 360 nm do not appear to follow exponential growth or decay during the time of the experiment. (The TTF graph at 308 nm is an exception). It is not appropriate to extract and report rate constants from exponential rate constant. The entire table with rate constants should be discarded. Is sufficient to explain that TTF degrades much more readily than DMPZ as evidenced in Fig 1.*

Also in response to your comment A) we removed this table and state as you suggested simply that TTF decays much more rapidly than DMPZ and also that the reduced form of each mediator decays more rapidly than the oxidized form.

J) *In line 141, you state “we oxidized the mediators electrochemically”, How exactly was this done- electrodes used? Concentration? Electrolyte solution, etc? What is the counterion to the radical cation (DMPZ⁺, TTF⁺) . Furthermore there is no indication as to the concentration of these radical cations in solution.*

The working electrode was a carbon paper of the type Freudenberg H2315, the glass fiber separator of the type Whatman GF/F. 50 μ L electrolyte (0.02 M RM (DMPZ or TTF) and 0.1 M LiTFSI in TEGDME) were used and cells were charged at 100 μ A to 3.5 and 3.7 V for DMPZ and TTF, respectively. Charge curves are shown below in Fig. R2 (the new Supplementary Fig. 4). The reached capacity was \sim 0.029 mAh, which closely matches the expected capacity for 1 e⁻ oxidation. The concentration of the oxidized mediator was therefore \sim 0.02 M. Since we charged the RMs in LiTFSI solutions, the counter ions for the DMPZ⁺ and TTF⁺ were TFSI⁻.

Figure R2. Charge curves for the oxidation of the RMs to RM^+ . 50 μ L electrolyte (0.02 M RM (DMPZ or TTF) and 0.1 M LiTFSI in TEGDME) were charged at 100 μ A in a Swagelok-type cell using a $Li_{1-x}FePO_4$ counter electrode and a carbon paper working electrode.

We revised the text on page 8 and Method section to make all this clear.

K) Line 150-151 “the DMPZ/DMPZ⁺ redox couple is degraded by IO_2 , because DMPZ reaction strongly, which degrades the DMPZ/DMPZ⁺ couple as a whole” This is tautological sentence. The same thing is said twice by saying the same thing in two different ways.

Thank you for pointing out this linguistic problem. We fixed it now.

L) The structure for DMPZ⁺ is presumably a radical cation such as shown for TTF⁺ and is drawn incorrectly in Fig3a.

The Reviewer is right that it must be a radical cation. We changed it at all occurrences to the structure as shown below.

M) Again are the data for Fig 3b, and Fig 3d continuous or discrete? Marker points should be used to indicate specific data points—there should not be just a simple line.

Values were not taken continuously but only at the discrete points indicated in Fig. 3a and c. To make this clear, we added markers in Fig. 3b and d to make this clear.

N) Line 169 should “peroxides” be replaced with “an organic peroxide” or even “peroxide”.

We have changed the text to “an organic peroxide”.

O) Line 173, there is an extra space “and TTF/TTF⁺, respectively,”.

Thank you, the mistake was corrected.

P) Lines 175, “We also carried out density functional (DFT)...” should be written as “We also carried out density functional theory (DFT)...”.

Thank you, the mistake was corrected.

Q) Line 190 “(Ref. 50)” should be “50”.

Thank you, we updated it to the common notation in this journal for references after subscript expressions, i.e., (ref. 50).

R) Fig 4- Species 4 is drawn is not drawn as a radical cation. Also it is a labeled species but was not labeled earlier in Fig 3a.

We changed DMPZ^+ at all occurrences to the radical cation. We have labelled also the RM and RM+ in Fig. 1 and 3 in a consistent manner to Fig. 4 and 5.

S) Fig 5, Species 13 appears to be a rearranged product where one of the 5 membered rings has shifted to a 6 membered ring. To form some kind of sulfoxide and ketone(?). There is no/little discussion in the text.

Species 13 is formed by rearrangement of a 5 membered ring to become a 6 membered ring with rearrangement and formation of a ketone in the ring. There is a large barrier and is not likely to occur.

T) Fig 5, There should be a ChemDraw for each molecule presented—it is confusing when you have only one diagram that supposedly represents multiple species (7, 8, 9, and 10) (and 12, 13, 14, and 15).

We have corrected the ChemDraw structures that did not correspond to the computational structures. Please note that it is difficult to exactly represent the computed structures in ChemDraw because there are some bonds that are between single/double and electrons distribution are not all integer.

U) Line 224, “The possibility for these products is supported by the NMR results in supplementary Fig 3b, which can be correlated to the formation of such products at one or multiple S-atoms. “.

This is a rather weak statement, all that is shown in Figure 3b is that the TTF has decomposed or degraded as a consequence of the reaction conditions. The NMR does not give indication as to the nature or identify of the products. If anything, the fact that the spectrum is so simple indicates that your degradation products may be represented by 2 or 3 major compounds and that a reasonable chance of isolation of the degradation products exists.

I would suggest the use of ESI-mass spectroscopy on the crude mixture to obtain the molecular weight of the unidentified products. Gas chromatography (with a mass spectrometer) would likely further degrade these presumed organic peroxides. The employment preparative (silica gel) TLC likely isolate and fully characterize these degradation products. This would allow you to focus your computation efforts on specific identified products.

We completely agree that what the NMRs show is degradation into multiple products. As common for oxidative decomposition reactions, rarely defined products are found². It is also typical for decomposition reactions that the onset step is typically the most demanding. The computed species represent likely the products of the onset reactions and will further decompose into a plethora of possible products.

Fig. R3 shows the new ¹H-NMR spectra of TTF before and after exposure to ¹O₂. Please note that the integral of all the new products are much less than the TTF at the start. The same is true for DMPZ decomposition. This means that the products still visible in the NMR do not represent all products the mediators are decomposing to. The most important step in degradation is the onset for which the computational results give important insights. The clear link between the computational and experimental work is the well reproduced trend of degradation rates of DMPZ vs. TTF and the reduced vs. the oxidized forms. This demonstrates that the found onset reactions are representative for the actually occurring ones. Therefore, even isolating and identifying a product, if at all possible, would not give a clearer indication of onset reactions.

Figure R3. (new Fig. S3b) Stability of TTF in contact with ¹O₂. ¹H-NMR spectra (in DMSO-d₆) were recorded before and after photooxygenation. 1 mg TTF in DME were stirred under an O₂ headspace. For photochemical ¹O₂ generation, 1 μM of the photosensitizer palladium(II) *meso*-tetra(4-fluorophenyl)-tetrabenzoporphyrin (Pd₄F) was dissolved in the solution and illuminated with 643 nm radiation. The DMSO peak is taken as internal reference for quantitative comparison of spectra.

V) Line 257 missing comma “(18-crown-6 >99%)”.

Thank you, the mistake was corrected.

² See, e.g., Curran et al. *Combust. Flame* 114, 149, (1998).

Electrochemical methods

A) *Line 266-269 “ The electrochemical cells used to oxidize the RMs were based on a Swagelok design....” This experimental description is vague. I have no idea what “Swagelok design” means other than you bought parts from a company—is there a better reference which explains your cell design. Also what kind of carbon cathode was use? What kind of glass fiber separator was used? What are the dimensions? The capacities? How much electrolyte was used?.*

These types of cells are widely used in battery research and the typical design of a three-electrode cell, as also used here, is given in Klink, S. et al. *Electrochem. Commun.* 22, 120-123, (2012). Our cell was based on a 1/2 inch Swagelok T-connector body and uses 12 mm diameter electrodes. The working electrode was a carbon paper was of the type Freudenberg H2315, the glass fiber separator of the type Whatman GF/F. 50 μ L electrolyte (0.02 M RM (DMPZ or TTF) and 0.1 M LiTFSI in TEGDME) were used and cells were charged at 100 μ A to 3.5 and 3.7 V for DMPZ and TTF, respectively. Charge curves are shown below in Fig. R1 (the new Supplementary Fig. 4). The reached capacity was \sim 0.32 mAh, which closely matches the expected capacity for 1 e⁻ oxidation.

We revised the Method section to make all this clear.

B) *Lined 275- “After the oxidation of the RM, the cells were disassembled immediately in the glovebox... extracted for the following experiments...” Into what were they extracted into? What solvent? At what concentration?*

The cell components (electrodes and separator) were immersed into 3.950 mL TEGDME to extract the electrolyte into a total of 4 mL TEGDME. The extract had thus a maximum concentration of 250 μ M RM⁺, but likely somewhat less since the extraction from the porous media will be slow. 1 μ M sensitizer was added for the following experiments with ¹O₂.

We revised the Method section to make all this clear.

C) *Furthermore, you mention the purification of LiTFSI but nowhere in the manuscript or experimental do you mention its use or its concentration. Presumably it was used as the electrolyte species in a solvent but at what concentration?*

We used LiTFSI as the electrolyte salt for oxidizing the mediators to their oxidized form. LiTFSI was used as 0.1 M solution in the TEGDME electrolyte. This point is clarified in the updated Methods section.

D) *Was TEGDME used as a solvent? There isn't any mention of its use in your experimental writeup.*

Yes, TEGDME was used as solvent in all UV-Vis and electrochemical experiments. In the revised manuscript we used dimethoxyethane (DME) for the new NMR measurements. DME was used to be able to evaporate the major part of the solvent prior to ¹H-NMR measurements and to thus get much more suitable RM concentrations in the NMR sample as before. We updated the Methods section to clarify in all cases what solvent was used.

E) *Was acetonitrile ever used as a solvent? There isn't any mention of its use in your experimental writeup or anywhere in the paper for that matter.*

Thank you for pointing this out. We haven't used acetonitrile anywhere and removed it from the Methods section.

F) Line 269 "(DMPZ and TTF)" should be "(DMPZ or TTF)".

Thank you, the mistake was corrected.

UV Vis and NMR analysis

A) Line 281 "were added to 2 mL of the RM-containing electrolytes and diluted before measurement using purified solvents". What solvents were used acetonitrile or TEGDME? What was the concentration of the RM mediator? Was LiTFSI used in the electrolyte? If so at what concentration and at what concentration?

Thank you. In the revised Methods section we clarified all sample preparations and also added the salt concentration to the Figure legends of Fig. 1 and 3. Electrolytes were TEGDME containing 0.1 M LiTFSI and 60 μ M RM.

B) What was the concentration of the sensitizer used?

In all cases we used 1 μ M sensitizer. We clarified this in the Figure legends of Fig. 1 and 3 and the Methods section.

C) What is OSRAM? Be nice to know if this is the company for the diode and what is the model number?

Yes. It is the name of company light-emitting diode light source. So, we inform the information like below. "the sensitizer was irradiated with a red light-emitting diode light source (OSRAM, 643 nm, 7 W)"

D) How were the lithium peroxide experiments carried out? There are indications in the main text of the paper but explicit details should be given in this section.

Thank you for pointing at this. 1 mg Li₂O₂ were stirred in 2 mL TEGDME containing 0.1 M LiTFSI and 60 μ M RM. We have revised the Methods section accordingly.

E) How were the oxygen experiments carried out? Was a solution made and allowed to sit on the benchtop? Was oxygen bubbled throughout the solution? Was the solution put in a vessel with an O₂ atmosphere, sealed and stirred for 24 h?

For measuring O₂ stability, 2 mL of the TEGDME electrolyte containing 0.1 M LiTFSI and 60 μ M RM were saturated with a stream of pure O₂ via a septum for 10 min and then the solution further stirred for 24 h in a closed 20 mL vial with pure O₂ headspace. We have revised the Methods section accordingly.

Reference section

A) Line 297 - Reference 1 has a missing author.

Thank you for pointing at the mistake. We have rechecked all citations.

B) On Line 317- only one page number is listed “1400867”; is this the correct format? The same can be said for Line 346, 351 370,377, 353, and 405 most references are cited with the range of page number- for example “*J. Phys. Chem. Lett. , 2989-2993.*”

Thank you for the close inspection. Some journals like the Adv. Mat. family, Nat. Energy, Nat. Comms. only use article numbers rather than page numbers. The mentioned references all belong to those journals. We have checked that all citations are correct.

C) Most of the references give the full authorship list. Many of the other references give a single author- for example “Mahne, et al” Is this the correct format?

The citation format in this journal is that up to six authors are listed in full, beyond six authors only the first is listed followed by “et al.”. We have checked that all citations are correct.

D) On Line 340, there is no space after the comma “IMLB,Jeju, Korea, S6-3 (2012).”.

Thank you, we corrected the mistake.

E) On Line 384, there is a typo “L-Air Batteries”.

Thank you, we corrected the mistake.

F) On Line 399, the full authors names are given “Last Name, First Name” instead of the normal abbreviation “Last Name, FN”.

Thank you for pointing at the mistake.

G) On Line 428, The “n” for “n-heptane” should be in italics?.

Thank you, we corrected the mistake.

H) On Line 437, there is an extra space “Density functional”.

Thank you, we corrected the mistake.

Supporting Information

A) Figure 1a --- “DMPZ in G4 baseline MeCN” What does this mean? What is G4- do you mean TEGDME? Why is MeCN mentioned? Your caption indicates TEGDME is the solvent. You should either use G4 or TEGDME not both. The method by which you expose your mediators to lithium peroxide is described in your caption and to some degree in your text but is absent in your experimental section. Presumably when you present your data the spectra indicating exposure to lithium peroxide after 24h is offset by 0.3 absorbance units. If this is the case, then the number scale should be deleted since your absorbance values for the offset spectra are arbitrary. Your caption should also make mention of this offset..

The label “baseline MeCN” was wrong and we removed it. G4 did mean TEGDME and we use the latter notation now throughout. We also revised the Methods section to clarify how all the experiments

were done in detail. The Reviewer is right that the spectra are offset for better visibility. We deleted the scale and mention the offset in the caption.

B) Figure 2: *What kind of NMR spectra is this? It should say " ¹H-NMR " in the caption. Your caption gives experimental details that appear at variance to your experimental details given in your experimental section. Was the procedure different for Uv-Vis KO₂/Li₂O₂ experiments compared to NMR experiments KO₂/Li₂O₂.*

What was the NMR solvent? Is it CDCl₃/TEGDME or CDCl₃ or TEGDME? You are missing quite a few peaks in your spectra---

If you used CDCl₃ as your NMR solvent there should be a CHCl₃ peak around 7.26 ppm. It seems that the spectra have been purposely cut at ~7.2 ppm.

Where are the aromatic resonances attributable to DMPZ at around ~7 ppm? Where are the methyl resonances (~3 ppm) attributable to DMPZ? Have they been cut off? Looking at Figure 2a there doesn't appear to be any DMPZ in your spectrum.

If you used TEGDME as indicated in the caption where are the resonances attributable to the ether?

What is this broad resonance around 6.5 to 6.0 ppm? Is it protonated TFSI salt?

In Figure 2b, where are the peaks attributable to CHCl₃ or TEGDME? You should show the full spectra? If you wish to show only the spectrum from 4.0 ppm to 7.2 ppm you can do that with an inset or a second or third figure

Thank you for the close inspection of the NMR part. In response to your insightful comments we completely redid all ¹H-NMR analysis. Previous spectra were obtained by using the same 60 μM mediator solutions in 0.1 M LiTFSI in TEGDME as used for UV-Vis analysis in the main paper and by dissolving them in CDCl₃. The low concentration of mediator in combination with the far overwhelming amount of TEGDME and the CDCl₃ caused all the problems you mentioned. E.g. we had previously cut the spectra to 4–7.2 ppm since the TEGDME and CDCl₃ peaks would by far dominate the spectra.

We thus redid all ¹H-NMR analysis with the following changes: (i) We use now 1.25 mg mediator per mL NMR solvent to have a much better signal-to-noise ratio. (ii) At this concentration it became apparent that CDCl₃ is not suitable since particularly the DMPZ spectra show extremely broad peaks; this was the reason why DMPZ in the previous Fig. S2a has shown only one extremely broad peak (the broad resonance around 6.5 to 6 ppm you mentioned). We thus used DMSO-d₆ instead, which allows for well resolved peaks. (iii) Instead of TEGDME we used dimethoxyethane (DME), which we removed by evaporation and thus allows for the higher mediator concentration in the NMR sample. All details of the procedures are mentioned in the updated Methods section.

C) Figure 3: *Again there should be an indication that this is a proton spectra. Your experimental section indicates that CDCl₃ was used but there is no indication in the caption.*

In the caption is the first mention of the concentration of the sensitizer. (1 μM).

I would consider putting a reference spectra for the sensitizer since it would assist the reader in peak identification (or lack of peaks).

So at what wavelength was the sample irradiated at? The text mentions 695 nm. Your experimental section mentions 643 nm. The caption in your supplementary indicates 625nm.

Figure 3a There appears to be negligible change in the spectra. This result is very different that your UV-Vis experiments would indicate. In Figure 2 of your main text, you assigned 342 nm as an absorbance for DMPZ. Over the course of 3 h the absorbance value for this wavelength dropped from 1.0 to 0.8. This would indicate a change of 20 % in DMPZ concentration yet there doesn't appear to a corresponding change in the NMR spectra. Admittely there is no internal standard for your NMR spectra, however there doesn't even appear to the growth of any new peaks in your NMR spectra consistent with related degradation products. Where is the missing 20 %? This discrepancy should be explained somewhere in your manuscript

Thank you for pointing at required improvements in the experimental description. We indicated now both in the experimental section and the figure captions of the NMRs that DMSO-d₆ was used as NMR solvent. Also the sensitizer concentration is mentioned both in the captions and the experimental part. At the concentration of 1 μM the sensitizer is entirely invisible in the NMR spectra. We apologize for the confusion with wavelengths. The correct value is 643 nm and is updated at all places where it is mentioned.

With regard to the change in the ¹H-NMR spectra of DMPZ before and after exposure to ¹O₂ we refer to Fig. R1 (the new Fig. S3a) which shows a decrease of DMPZ concentration in accord with the UV-Vis spectra in the main text. We used the DMSO peak as internal standard for quantitative comparison of the spectra. Please note that the integral of all the new products are much less than the DMPZ at the start. This means that the products still visible in the NMR do not represent all products the mediators are decomposing to. They may form inorganic products or evolve as gases. We explain this now in the manuscript on page 5.:

“Of note, the integral of all the new products are much less than the mediators at the start. This means that the products still visible in the NMR do not represent all products the mediators are decomposing to. They may form inorganic products or evolve as gases.”

Reviewer #2:

General impression

Freunberger and Sun et al. reported on the deactivation of redox mediator (RM) for lithium-oxygen cells caused by singlet oxygen, via a combined experimental and computational approach. This cutting-edge research is of utmost importance to the field of metal-oxygen batteries, as it identifies the culprit of RM degradation, which is a long-standing puzzle, and therefore points out a promising pathway to diminish or radically eliminate RM degradation. The manuscript is clearly written. The major arguments are well grounded on experimental facts. Considering these points, this reviewer is

happy to recommend publication of this work after minor revisions that I believe will further improve the quality of this work.

We thank the reviewer for the very favourable assessment of our work and for pointing at the implications for the further development of Li-O₂ batteries.

Two suggestions

(1) Further discussion of the implications and their demonstration in a lithium-oxygen cell

At its present state, this work mainly compares the reactivity of O₂, superoxide and singlet oxygen towards two RMs and then dwells upon the reaction mechanism. Discussion on the implication of this study to lithium-oxygen cells is refrained. An important missing part to collaborate the conclusion that singlet oxygen is the major culprit of RM degradation is to demonstrate that by somehow trapping singlet oxygen the lithium-oxygen battery with RM has a longer lifetime.

Thank you for this helpful suggestion. We have extended the Discussion section with implications for the future of research in Li-O₂ cells. They range from a more general paradigm shift in how stability of materials should be viewed via possibilities to improve mediators through experimental and computational work towards nevertheless required measures to counteract ¹O₂ formation.

After having clarified in this work that mediators are predominantly degraded by ¹O₂ and how possible onset reactions cause the degradation we feel that to comprehensively show how traps or quenchers could improve the lifetime of batteries would require a full study on its own right. This is ongoing work in our group and will be reported in the future. Overall, we think this study on mediator decomposition by ¹O₂ will play a critical role as a platform for further improvement.

(2) Reorganization of the results section and extension of the discussion section

The presented manuscript has a very short discussion section. I found that extensive discussion is actually contained in the results section. I would suggest the authors to move those parts to the discussion section.

We were thinking hard about what part of the discussion could be moved from the Results section (actually as you noted rather a combined Results and Discussion section) to the Discussion section without impeding the flow of thoughts and helping understanding. We have arranged with combined Results and Discussion as we thought it is easier to understand since later presented data build upon the insights of earlier presented ones.

We now extended the discussion section to more widely discuss the implications of the study on future research in Li-O₂ cells as you suggested in the above comment. We hope that this way we could satisfyingly respond to both of your comments and that we overall present the paper in a clear way with a clear understanding of the wider implications.

Reviewers' comments:

Reviewer #1 (Remarks to the Author):

Summary

This manuscript is not ready for publication. Furthermore, there are some problems with the NMR data as presented in the manuscript/supporting information. These problems and outlines in the “General Notes” and specific issues are covered line-by-line below.

General Notes

- A) Labels for molecules should be labeled in the same order in which they are introduced:
- DMPZ is the first labeled molecule introduced in Figure 1 and is given the label “1”.
 - TTF is the second labeled molecule introduced in Figure 2 and is given the label “6”
 - DMPZ+ is the third labeled molecule introduced in Figure 3 and is given the label “4”
 - TTF+ is the fourth labeled molecule introduced in Figure 4 and is given the label “16”
 - The number labels appear to be assigned on the basis of the computational work which is near the end of the manuscript
- B) Abbreviations for chemical species are used inconsistently throughout. The species dissolved oxygen (O₂), superoxide (KO₂), peroxide (Li₂O₂), and singlet oxygen (1O₂) are used throughout the manuscript- sometimes by abbreviation and sometimes by name. There does not appear to be a consistent standard. Examples are listed (see below).
- C) One of the points raised in my initial review was not adequately addressed: In the previous review, I made following statement (emphasis in red added) that

Line 266-269 “ The electrochemical cells used to oxidize the RMs were based on a Swagelok design....” **This experimental description is vague. I have no idea what “Swagelok design” means other than you bought parts from a company—is there a better reference which explains your cell design.** Also what kind of carbon cathode was used? What kind of glass fiber separator was used? What are the dimensions? The capacities? How much electrolyte was used?.

The response/rebuttal was as follows (emphasis in red added)

These types of cells are widely used in battery research and the typical design of a three-electrode cell, as also used here, is given in Klink, S. et al. Electrochem. Commun. 22, 120-123, (2012). Our cell was based on a 1/2 inch Swagelok T-connector body and uses 12 mm diameter electrodes. The working electrode was a carbon paper was of the type Freudenberg H2315, the glass fiber separator of the type Whatman GF/F. 50 μ L electrolyte (0.02 M RM (DMPZ or TTF) and 0.1 M LiTFSI in TEGDME) were used and cells were charged at 100 μ A to 3.5 and 3.7 V for DMPZ and TTF, respectively. Charge curves are shown below in Fig. R1 (the new Supplementary Fig. 4). The reached capacity was ~0.32 mAh, which closely matches the expected capacity for 1 e⁻ oxidation.

Upon reading the methods section (and manuscript), I could not find the reference [Klink, S. et al. **Electrochem. Commun. 22, 120-123, (2012).**] that described the swagelok cell dimensions or its design as described in the rebuttal. The reader should be able to determine what is meant by a “swagelok

cell”.

- D) In the previous review, I suggested that the computational work was somewhat disconnected from the experimental work and that perhaps an attempt could be made to isolate one of the degradation products to narrow the computational work.

The rebuttal stated:

As common for oxidative decomposition reactions, rarely defined products are found¹. It is also typical for decomposition reactions that the onset step is typically the most demanding. The computed species represent likely the products of the onset reactions and will further decompose into a plethora of possible products. Therefore, even isolating a product, if at all possible, would not give a clearer indication of onset reactions.

[Footnote: 1 See, e.g., Curran et al. Combust. Flame 114, 149, (1998)]

First, the cited paper concerns the combustion of n-heptane with oxygen which yields the well-defined products water and carbon dioxide. Furthermore, This particular paper is not germane since you are concerned with incomplete oxidation at room temperature of your organic candidates using the singlet oxygen reaction. These reaction conditions for alkane combustion are wildly different.

Second, there are several examples of the synthetic utility of singlet oxygen. For some relevant examples employing singlet oxygen see Chem Rev. 1981(81) 91-108, Org Lett 2007, 5585, and J. Org. Chem. 2004, 7875. The manuscript that you wrote also seems to reference (ref #'s 48, 49, 51, 55, 56, and 57) several examples of singlet oxygen reactions that do result in well-defined species.

Finally, your own paper makes the following statement “The possibility for these products [12,13,14,15 --- organic sulfones/sulfoxide] is supported by the NMR results in Supplementary Figure 3b, which can be correlated to the formation of such products at one or multiple S-atoms”

If you are claiming that NMR shows a plethora of reaction products that likely cannot be isolated or identified (as stated in your rebuttal- [Therefore, even isolating and identifying a product, if at all possible, would not give a clearer indication of onset reactions]), then NMR cannot make any kind of comment on the possibility of organic sulfones or sulfoxides. There is no logical connection between the NMR data and the computational work.

E) Again the NMR work is problematic. Solvent peaks are often used to reference spectra for example the chloroform peak in CDCl_3 is taken is typically taken to be at 7.26 ppm. ***Solvent peaks are not used for quantification since the purity of the solvent peaks varies from bottle to bottle and will vary even between samples as it absorbs water from the environment which can exchange protons with the deuterated solvent.***

Furthermore, since the actual concentration of the solvent peak is unknown (the purity given on the bottle is merely a lower limit) then the concentration of the compound being studied also cannot be known.

Quantification for NMR is typically done using either a *calibrated* sealed capillary (external standard) containing a known quantity of compound (e.g. benzene) or by adding a known amount of a non-volatile compound to the NMR solution (internal standard). Since the concentration/quantity of the added compound is now known. Its integrated NMR signal can now be compared to the analytes signal to quantitatively determine its concentration.

For an example see

<http://www.organ.su.se/bo/Gruppfiler/NMR%20yield%20calculation.pdf>

Also since quantification by NMR is an important portion of your manuscript, It would be most helpful for the quantification/yield to be given in the supporting information. For example in Figure 2b of the Supporting information, does the measured TTF represent 90% of the expected value or 100 % or 5 %? ***I noticed the ratio of water to your analyte seems to vary considerably in Figure 2 (the integrals are not listed) of the supporting information which indicates that your “solvent reference peak” is either not constant or the amount of TTF that you are measuring is highly variable.***

In your manuscript, Line 103-105, you claim to observe a 20 % decrease in DMPZ intensity for your NMR data. Since your NMR data appear to not have been quantitative, such a statement can not be made.

Manuscript

- A) Line 56--- what is meant thermodynamically stable RM's? do you mean kinetically stable? Or stable to lithium metal? A similar point was raised in the previous review (in a different section of the manuscript—see line 83) about how the statement “thermodynamically stable RM” was vague.
- B) Line 56 – replace “Li metal” with “lithium metal.” The phrase “lithium metal” is used on line 52. Pick a consistent standard.
- C) Line 65- 67- is a runon sentence which is difficult to understand.

Here we assess the reactivity of organic RM's towards dissolved oxygen, superoxide, peroxide, and $^1\text{O}_2$ using quantitative UV-Vis analysis and NMR and demonstrate the predominant cause for RM deactivation to be $^1\text{O}_2$.

Could be written as :

Here we assess the reactivity of organic RM's towards dissolved oxygen (O_2), potassium superoxide (KO_2), lithium peroxide (Li_2O_2), and $^1\text{O}_2$ using quantitative UV-Vis analysis and $^1\text{H-NMR}$. We demonstrate the predominant cause for RM deactivation to be $^1\text{O}_2$.

- D) Line 65- as suggested above, the word “peroxide” should be replaced with “lithium peroxide (Li_2O_2)” to match what is written in Line 78 as “ Li_2O_2 ”
- E) Line 65- as suggested above, “dissolved oxygen” should be written as “dissolved oxygen (O_2)” to match the abbreviation “ O_2 ” you use in line 78
- F) Line 79- replace “Singlet oxygen” with “ $^1\text{O}_2$ ” since you introduced the abbreviation in in line 62. Once an abbreviation/label/acronym is introduced in the manuscript it should be used consistently throughout the manuscript. Do not go back and forth- pick a consistent standard.
- G) Line 78 (or Line 65-67 as suggested above) should introduce abbreviation potassium superoxide (KO_2) since you use this (KO_2) abbreviation in line 87.
- H) Line 70 – replace “S” with “sulfur”
- I) Line 85- I would suggest the following as easier to read: **The RMs were dissolved in tetraethylene glycol dimethyl ether (TEGDME) at a concentration (60 uM) suitable for UV-Vis spectroscopy.**
- J) Line 89- is the amount/concentration of crown ether given in the experimental section
- K) Line 89 – since you are starting a sentence it looks cleaner by saying “Dissolved oxygen (O_2)” instead of the abbreviation (O_2)
- L) Line 93- “These results correspond” should be “These results are consistent”
- M) Line 93- use the abbreviation KO_2 instead of superoxide
- N) Line 108- how does one form inorganic products from organic products? The structures shown in figure 4 are not inorganic or gases

“They may form inorganic products or evolve as gases. Which are shown in Fig. 4 and will be discussed later. “

- O) Line 109- there is a punctuation error
- P) Line 132 – Since you starting a sentence use “Singlet oxygen ($^1\text{O}_2$)” instead of just “Singlet oxygen”
- Q) Line 133 – This is nearly a tautological statement:

Singlet oxygen is known to react with electron-rich organic substrates containing C=C double bonds via so-called “ene” or “diene” reactions driven by the electrophilic nature of $^1\text{O}_2$.

The first part of the sentence implies that singlet oxygen is electrophilic by using the phrase “electron-rich” and then reiterates this point by describing it as electrophilic. Consider instead:

“Singlet oxygen ($^1\text{O}_2$) is known to react with electron-rich carbon-carbon double bonds via so-called “ene” or “diene” reactions.”

- R) Line 137- the phrase “towards $^1\text{O}_2$ ” is superfluous
- S) Line 144 – “Fig. 2 a” should be written as “Fig 2a”
- T) Line 153-154—There is no more table 1—the sentence referring to table 1 should be deleted
- U) Line 156- again you are not consistent with words vs abbreviations.

In all cases, the reaction with $^1\text{O}_2$ clearly dominates the possible reactions with other oxygen species, O_2 , superoxide, and peroxide

The above sentence implies that there are other oxygen species besides ‘ O_2 , superoxide, and peroxide’ The sentence should be written as

In all cases, the reaction with $^1\text{O}_2$ clearly dominates the other possible reactions with O_2 , KO_2 , and Li_2O_2

- V) Line 157- Figure 3- the structure for the radical cation form of DMPZ is drawn incorrectly- one carbon drawn as a radical is implied to exceed the octet rule and another carbon has an implied hydrogen where none exists
- W) Line 160- The caption sentence is written in such a way to imply that you are oxidizing the electrolyte solvent when you really mean to oxidize the DMPZ.
- X) Line 176 – the two abbreviations are not consistent
 - a. Is B3LYP/6-31+G* supposed to be the same as B3LYP/6-31G*
 - b. See also line 202 and line 180
- Y) Line 189- the reference is not written correctly “(ref. 50)” should be “⁵⁰”
- Z) Line 231- since Table 1 was deleted why is it being referenced?

AA) Line 239-Line 252 The newly discussion section could use a little revision-> see comments in red

The results of this study have multiple implications for required research directions in Li-O₂ cells towards developing practical energy storage devices. First, generally, the previous paradigm that stability of cell components against superoxide [abbreviation?] and peroxide [abbreviation?] were of prime importance needs to shift towards additionally and even more importantly stability against ¹O₂. This concerns both studies on how materials degrade and on making more stable materials. Second, redox mediation is now widely accepted to be key for Li-O₂ batteries to achieve maximum energy density and efficiency by far higher rates than possible without [this is rather awkward phrasing -what is meant by “higher rates” – a rate does not describe energy density or efficient?] The fact that we have shown that mediators can have very different susceptibility to decompose with ¹O₂ spurs hope that even more stable mediators will be found. Third, the computational results that nicely reproduce the trend [this study only considers two different mediators!] in reactivity between different mediators and between the reduced and oxidized states suggest that computational screening is a very [since you didn't use computation to screen or preselect candidate mediator and you only investigates two mediator how can you make the claim that computational screening is an effective tool for screening--- in fact you stated that you chose DMPZ for the low overpotential and TTF as one of the first reported RM's] effective tool to preselect candidate mediators. Fourth, even the best accessible [what is meant by accessible?] mediators may not be sufficiently stable for long term operation in presence of ¹O₂. Therefore additional means of counteracting degradation will likely be required. These may be chemical traps that more rapidly react with ¹O₂ than other cell components, or, preferably, physical quenchers.

BB) Line 252- what is physical quencher?- presumably you mean a molecule that can act to catalytically convert singlet oxygen to triplet oxygen- a little explanation here would help the reader

CC) Line 256- “Li “should read “lithium”

DD) Line 259 – as previously mentioned, thermodynamically is not well defined in this statement--- see line 83 where a revision was made

Experimental

EE) Line 280- again what is a Swagelok design? your rebuttal gave a reference where presumably the construction of the cell is described but again the details of this cell are absent from your manuscript and cannot be found in the supplementary information

FF) Line 316- solvent peaks are typically used to reference spectra for example the chloroform peak in CDCl₃ is typically taken to be at 7.26 ppm. Solvent peaks are not used for quantification since the purity of the solvent peaks varies from bottle to bottle and will vary even between samples as it absorbs water from the environment.

References

GG) Line 390- There are characters that do not appear to render correctly in the copy of the PDF that I have obtained. This may be a problem during publication. If it appears correct on your computer you may want to confirm with the editor that it will render correctly. Similar problems are seen at lines 390, 393, and 457.

HH) For your references "doi" links are given. Is this the correct format for this journal? I downloaded to recent nature communication papers and it does not seem that this is the case.

II) Line 392- the phrase "doi" is repeated

JJ) Line 395- the phrase "doi" is repeated

KK) Line 397: you have the phrase "na/na" in the reference

LL) Line 448: the phrase "doi" is repeated

Supplementary Information:

Line 37: is "photo-oxygenation" misspelled?

Line 38: verb subject;" were" should be "was"

Figure 4: what is the diameter of the cell ?

Reviewer #2 (Remarks to the Author):

I can conclude that I am happy with the changes by the authors. The authors have addressed my concerns and I believe the manuscript can be published in its current form.

Response to reviewers on the paper entitled

“Deactivation of redox mediators in Li-O₂ batteries by singlet oxygen”

manuscript: NCOMMS-18-24458A

We thank the Reviewers for their helpful comments to which we respond below point by point. The Reviewers' comments are reproduced in italics. In the manuscript and supporting information we highlighted the changes in yellow. The comments helped us to greatly improve the manuscript and we are confident that we have satisfyingly responded to all of them.

Reviewer #1:

Summary

This manuscript is not ready for publication. Furthermore, there are some problems with the NMR data as presented in the manuscript/supporting information. These problems and outlines in the “General Notes” and specific issues are covered line-by-line below.

We thank the Reviewer for careful comments to our manuscript. We responded to all the comments point by point like below.

General Notes

A) Labels for molecules should be labeled in the same order in which they are introduced:

a. DMPZ is the first labeled molecule introduced in Figure 1 and is given the label “1”.

b. TTF is the second labeled molecule introduced in Figure 2 and is given the label “6”

c. DMPZ⁺ is the third labeled molecule introduced in Figure 3 and is given the label “4”

d. TTF⁺ is the fourth labeled molecule introduced in Figure 4 and is given the label “16”

e. The number labels appear to be assigned on the basis of the computational work which is near the end of the manuscript.

We have now renumbered all compounds to appear strictly in the order in which they are introduced. Changes are therefore to Fig. 1, 3, 4 and 5 and the text associated with it. Changes to the text are highlighted in yellow.

B) Abbreviations for chemical species are used inconsistently throughout. The species dissolved oxygen (O₂), superoxide (KO₂), peroxide (Li₂O₂), and singlet oxygen (1O₂) are used throughout the manuscript- sometimes by abbreviation and sometimes by name. There does not appear to be a consistent standard. Examples are listed (see below).

Thank you for helping us with this linguistic issue. We have now changed all abbreviations in accordance with the Reviewer's suggestions.

C) One of the points raised in my initial review was not adequately addressed: In the previous review, I made following statement (emphasis in red added) that

Line 266-269 “ The electrochemical cells used to oxidize the RMs were based on a Swagelok design....” *This experimental description is vague. I have no idea what “Swagelok design” means other than you bought parts from a company—is there a better reference which explains your cell design.* Also what kind of carbon cathode was use? What kind of glass fiber separator was used? What are the dimensions? The capacities? How much electrolyte was used?.

The response/rebuttal was as follows (emphasis in red added)

These types of cells are widely used in battery research and the typical design of a tree-electrode cell, as also used here, is given in Klink, S. et al. Electrochem. Commun. 22, 120-123, (2012). Our cell was based on a 1/2 inch Swagelok T-connector body and uses 12 mm diameter electrodes. The working electrode was a carbon paper was of the type Freudenberg H2315, the glass fiber separator of the type Whatman GF/F. 50 μ L electrolyte (0.02 M RM (DMPZ or TTF) and 0.1 M LiTFSI in TEGDME) were used and cells were charged at 100 μ A to 3.5 and 3.7 V for DMPZ and TTF, respectively. Charge curves are shown below in Fig. R1 (the new Supplementary Fig. 4). The reached capacity was \sim 0.32 mAh, which closely matches the expected capacity for 1 e⁻ oxidation.

Upon reading the methods section (and manuscript), I could not find the reference [Klink, S. et al. Electrochem. Commun. 22, 120-123, (2012).] that described the swagelok cell dimensions or its design as described in the rebuttal. The reader should be able to determine what is meant by a “Swagelok cell”.

We are sorry that our previous response was not clear enough. We now put a figure with cell dimensions and the part number of the used fitting from the company Swagelok as new Supplementary Figure 7. For your convenience we insert it below as Fig. R1. We updated the Methods Section accordingly.

Figure R1 | (new Supplementary Figure 7) The cell design used for electrochemically oxidizing the RM to RM^+ . The central piece is a union tee tube fitting from the company Swagelok (<https://www.swagelok.com/en/catalog/Product/Detail?part=PFA-820-3>) made from perfluoroalkoxy polymer (PFA). The plungers for counter, working and reference electrodes are made from stainless steel (grade SAE 316L).

D) In the previous review, I suggested that the computational work was somewhat disconnected from the experimental work and that perhaps an attempt could be made to isolate one of the degradation products to narrow the computational work.

The rebuttal stated:

As common for oxidative decomposition reactions, rarely defined products are found¹. It is also typical for decomposition reactions that the onset step is typically the most demanding. The computed species represent likely the products of the onset reactions and will further decompose into a plethora of possible products. Therefore, even isolating a product, if at all possible, would not give a clearer indication of onset reactions.

[Footnote: 1 See, e.g., Curran et al. *Combust. Flame* 114, 149, (1998)]

First, the cited paper concerns the combustion of n-heptane with oxygen which yields the welldefined products water and carbon dioxide. Furthermore, This particular paper is not germane since you are concerned with incomplete oxidation at room temperature of your organic candidates using the singlet oxygen reaction. These reaction conditions for alkane combustion are wildly different.

*Second, there are several examples of the synthetic utility of singlet oxygen. For some relevant examples employing singlet oxygen see *Chem Rev.* 1981(81) 91-108, *Org Lett* 2007, 5585, and *J. Org. Chem.* 2004, 7875. The manuscript that you wrote also seems to reference (ref #'s 48, 49, 51, 55, 56, and 57) several examples of singlet oxygen reactions that do result in well-defined species.*

Finally, your own paper makes the following statement “The possibility for these products [12,13,14,15 --- organic sulfones/sulfoxide] is supported by the NMR results in Supplementary Figure 3b, which can be correlated to the formation of such products at one or multiple S-atoms”

If you are claiming that NMR shows a plethora of reaction products that likely cannot be isolated or identified (as stated in your rebuttal- [Therefore, even isolating and identifying a product, if at all possible, would not give a clearer indication of onset reactions]), then NMR cannot make any kind of comment on the possibility of organic sulfones or sulfoxides. There is no logical connection between the NMR data and the computational work.

Regarding the connection of the computational and experimental work we want to stress what the point of the calculations is: they show the barriers for *onset reactions* as the first step of singlet O₂ attack on to mediators. It is also typical for decomposition reactions that the *onset step* is the most demanding. Hence calculations provide a rationale for the different stability of the mediators. As requested, we further could identify one of the degradation products (see below), which makes a clear connection between computational work and NMR analysis and further corroborates that the degradation reactions DO NOT lead to unique products, but rather to a plethora of different ones.

First: We agree that the reference may have been somewhat misleading. We wanted to argue, that some of the calculated onset products are known as the key intermediates in any oxidative decomposition (combustion). In particular R-OOH, R•, and R-OO• species, which once formed continue to break down in chain reactions with a multitude of reaction paths as stated in the reference. The same species are also common onset products in ene reactions with ¹O₂ (e.g., Ref. 49).

Second: Of course the Reviewer is right that ¹O₂ is used in organic synthesis. However, as common in organic synthesis often the product yield of the intended substance of the listed references is not quantitative and accompanied by significant side products. It will not have escaped the Reviewer’s attention that the synthetic works that use ¹O₂ as reagents typically use cryogenic conditions. For example, Ref. 49 at –80° C, Ref. 51 ice path, Ref. 56 at –78° C and so on. Such conditions are used in synthesis to suppress ongoing reactions of the product(s) with the reagents. Otherwise one ends up with a plethora of products and no well-defined products. Reactions of DMZP and TTF with ¹O₂ at room temperature are no different.

Finally: Thank you for this very valuable comment. We were now able to identify the major specific product of TTF decomposition, which confirms the validity of the computed reaction pathways. The attack of ¹O₂ on the central C=C bond to form the dioxetane product **12** is the most favorable of the ones shown. Dioxetanes are known to undergo decomposition to the related carbonyl compounds by cleaving the C–C bond (see e.g. *J. Am. Chem. Soc.* **98**, 1086 (1976)). Mechanisms reported involve either [2+2] cycloelimination or a radical mechanism (*Chem. Rev.* **118**, 6927 (2018)). In fact, even the sensitized photooxygenation of a related compound has directly been reported to undergo such cleavage. It is shown to form the corresponding dithiocarbonate (Fig. R2). The corresponding mechanism with TTF is shown in Fig. R3 to form 1,3-dithiol-2-one. It is clearly seen in the NMR as the major newly formed decomposition product with a singlet at 7.1 ppm. We updated Supplementary Fig. 3 where the NMR spectrum is shown (for your convenience reproduced below in Fig. R4). However, all the NMR visible products together after photooxygenation only equate to 40 % of the initial TTF of which 28% are 1,3-dithiol-2-one. Hence, 1,3-dithiol-2-one decomposes further with ¹O₂, likely via pathways resembling combustion reactions. There are minor peaks at ~5.8 and 1.2 ppm which we could not identify, but which are by far not accounting for all lost TTF.

Taken together, the identification of the specific product resulting from the most favorable computational path shows clearly that the computational work is indeed able to predict the likely pathways. Our computational work is hence a very valuable addition to the experiments. We have updated the text on page 13-14 accordingly.

Figure R2 | The published pathway of sensitized photooxygenation of 6,6-(Ethylenedithio)fulvene **9** to form the corresponding dithiocarbonate. From: Zhang, X., Lin, F. & Foote, C. S. Sensitized Photooxygenation of 6-Heteroatom-Substituted Fulvenes: Primary Products and Their Chemical Transformations. *J. Org. Chem.* **60**, 1333 (1995).

Figure R3 | (new Supplementary Figure 4) Decomposition of TTF to 1,3-dithiol-2-one according to the mechanism proposed by Zhang, X., Lin, F. & Foote, C. S. Sensitized Photooxygenation of 6-Heteroatom-Substituted Fulvenes: Primary Products and Their Chemical Transformations. *J. Org. Chem.* **60**, 1333 (1995)

Figure R4 | (the updated Supplementary Figure 3b) $^1\text{H-NMR}$ spectra (in DMSO-d_6) of TTF in DME before and after photooxygenation.

E) Again the NMR work is problematic. Solvent peaks are often used to reference spectra for example the chloroform peak in CDCl_3 is taken is typically taken to be at 7.26 ppm. **Solvent peaks are not used for quantification since the purity of the solvent peaks varies from bottle to bottle and will vary even between samples as it absorbs water from the environment which can exchange protons with the deuterated solvent.**

Furthermore, since the actual concentration of the solvent peak is unknown (the purity given on the bottle is merely a lower limit) then the concentration of the compound being studied also cannot be known.

Quantification for NMR is typically done using either a calibrated sealed capillary (external standard) containing a known quantity of compound (e.g. benzene) or by adding a known amount of a non-volatile compound to the NMR solution (internal standard). Since the concentration/quantity of the added compound is now known. Its integrated NMR signal can now be compared to the analytes signal to quantitatively determine its concentration.

For an example see

<http://www.organ.su.se/bo/Gruppfiler/NMR%20yield%20calculation.pdf>

Also since quantification by NMR is an important portion of your manuscript, It would be most helpful for the quantification/yield to be given in the supporting information. For example in Figure 2b of the Supporting information, does the measured TTF represent 90% of the expected value or 100 % or 5 %? I noticed the ratio of water to your analyte seems to vary considerably in Figure 2 (the integrals are noe listed) of the supporting information which indicates that your “solvent reference peak” is either not constant or the amount of TTF that you are measuring is highly variable.

In your manuscript, Line 103-105, you claim to observe a 20 % decrease in DMPZ intensity for your NMR data. Since your NMR data appear to not have been quantitative, such a statement can not be made.

We prepared all stability tests in parallel and furthermore prepared the samples for NMR spectroscopy with the very same deuterated solvent out of the same bottle (DMSO-d₆) within ~5 min. Hence, possible differences between bottles are excluded. Furthermore, DMSO is known to be very very mildly acidic. Considering that the pK_a values of water and DMSO differ by ~20 pK_a units, the exchange of protons of the water with deuterium atoms from DMSO-d₆ is extremely unlikely. For that reason, the relative amount of protons from DMPZ to DMSO can be used to relate the content of different samples. Knowing absolute values of undeuterated DMSO is not required. Hence, the values before and after photooxygenation can be used for quantitative comparison.

Values for the decrease of TTF in Supplementary Fig. 3b have been discussed above. For the DMPZ, relative amount of DMPZ compared to DMSO after 3 h ¹O₂ exposure is 0.1143 (see attached image; Supplementary Figure 5), hence ~62% of the initial value (0.1847). This decay of DMPZ is in accord with the UV-Vis measurement that shows a decay to ~65%.

Figure R5 | (new Supplementary Figure 5) Integrals of the ^1H -NMR spectra in Fig S3a.

Manuscript

A) Line 56--- what is meant thermodynamically stable RM's? do you mean kinetically stable? Or stable to lithium metal? A similar point was raised in the previous review (in a different section of the manuscript—see line 83) about how the statement “thermodynamically stable RM” was vague.

What the statement refers to is related to the paper by K. Kang et al. (ref. 22, *Nat. Energy* 1, 16066 (2016)). They have shown that the SOMO energy of the oxidized redox mediator must not be lower than the HOMO of the electrolyte solvent in order not to oxidize the solvent. We changed the text to: ...the SOMO energy of the oxidized mediator must not be lower than the HOMO of the electrolyte solvent²².

B) Line 56 – replace “Li metal” with “lithium metal.” The phrase “lithium metal” is used on line 52. Pick a consistent standard.

We revised all “Li metal” to “lithium metal”.

C) Line 65- 67- is a runon sentence which is difficult to understand.

Here we assess the reactivity of organic RM's towards dissolved oxygen, superoxide, peroxide, and IO_2 using quantitative UV-Vis analysis and NMR and demonstrate the predominant cause for RM deactivation to be IO_2 .

Could be written as :

Here we assess the reactivity of organic RM's towards dissolved oxygen (O₂), potassium superoxide (KO₂), lithium peroxide (Li₂O₂), and ¹O₂ using quantitative UV-Vis analysis and 1H-NMR. We demonstrate the predominant cause for RM deactivation to be ¹O₂.

Thanks for your recommendation. We revised this sentence according to your suggestion (Li₂O₂ is already mentioned before this sentence).

D) Line 65- as suggested above, the word "peroxide" should be replaced with "lithium peroxide (Li₂O₂)" to match what is written in Line 78 as "Li₂O₂"

We revised the first occurrence of peroxide to "lithium peroxide (Li₂O₂)" and write "Li₂O₂" thereafter.

E) Line 65- as suggested above, "dissolved oxygen" should be written as "dissolved oxygen (O₂)" to match the abbreviation "O₂" you use in line 78

We revised the first occurrence to "dissolved oxygen (O₂)" and write "O₂" thereafter.

F) Line 79- replace "Singlet oxygen" with "¹O₂" since you introduced the abbreviation in in line 62. Once and abbreviation/label/acronym is introduced in the manuscript it should be used consistently throughout the manuscript. Do not go back and forth- pick a consistent standard.

We checked that "¹O₂" is written throughout after the abbreviation is first introduced.

G) Line 78 (or Line 65-67 as suggested above) should introduce abbreviation potassium superoxide (KO₂) since you use this (KO₂) abbreviation in line 87.

Thanks for carefully checking the manuscript. We introduce abbreviation for potassium superoxide (KO₂) in line 65-66 and superoxide (O₂⁻) in line 60. KO₂ is used as a chemical O₂⁻ source. In the experiments where we expose the RM to superoxide we use crown ether solubilized KO₂, which results in dissolved O₂⁻. We carefully checked to use KO₂ and O₂⁻ as appropriate.

H) Line 70 – replace "S" with "sulfur"

Done.

I) Line 85- I would suggest the following as easier to read: The RMs were dissolved in tetraethylene glycol dimethyl ether (TEGDME) at a concentration (60 μM) suitable for UV-Vis spectroscopy.

Thank you for the helpful comment. We changed this sentence as suggested.

J) Line 89- is the amount/concentration of crown ether given in the experimental section

Yes, we mention the amount of crown ether in experimental part.

K) Line 89 – since you are starting a sentence it looks cleaner by saying "Dissolved oxygen (O₂)" instead of the abbreviation (O₂)

Thank you, we revised the sentence accordingly.

L) Line 93- "These results correspond" should be "These results are consistent"

We revised the sentence.

M) Line 93- use the abbreviation KO₂ instead of superoxide

In accordance with comment G) we changed it to O₂⁻, meaning dissolved superoxide.

N) Line 108- how does one form inorganic products from organic products? The structures shown in figure 4 are not inorganic or gases

“They may form inorganic products or evolve as gases. Which are shown in Fig. 4 and will be discussed later. “

As mentioned in comment D) combustion (or other oxidative decomposition passing via R-OOH, R•, and R-OO• species) eventually decomposes organic matter to H₂O and CO₂ (or S or N containing gases, salts, or the like if the matter contained them), once the barrier for the onset steps is overcome. They are all inorganic and invisible to NMR and at least some of them gases. We change it to

They may form inorganic products or evolve as gases **as ultimate products of oxidative decomposition reactions as, which are shown in Fig. 4 and will be** discussed later.

O) Line 109- there is a punctuation error

Thank you. We revised it.

P) Line 132 – Since you starting a sentence use “Singlet oxygen (1O₂)” instead of just “Singlet oxygen”

Done.

Q) Line 133 – This is nearly a tautological statement: Singlet oxygen is known to react with electron-rich organic substrates containing C=C double bonds via so-called “ene” or “diene” reactions driven by the electrophilic nature of 1O₂.

The first part of the sentence implies that singlet oxygen is electrophilic by using the phrase “electron-rich” and then reiterates this point by describing it as electrophilic.

Consider instead:

“Singlet oxygen (1O₂) is known to react with electron-rich carbon-carbon double bonds via so-called “ene” or “diene” reactions.”

Thank you, we changed the sentence as suggested.

R) Line 137- the phrase “towards 1O₂” is superfluous

We removed it.

S) Line 144 – “Fig. 2 a” should be written as “Fig 2a”

We revised it to “**Fig. 2a**”.

T) Line 153-154—There is no more table 1—the sentence referring to table 1 should be deleted

We removed this sentence.

U) Line 156- again you are not consistent with words vs abbreviations.

In all cases, the reaction with IO_2 clearly dominates the possible reactions with other oxygen species, O_2 , superoxide, and peroxide

The above sentence implies that there are other oxygen species besides ‘ O_2 , superoxide, and peroxide’ The sentence should be written as

In all cases, the reaction with IO_2 clearly dominates the other possible reactions with O_2 , KO_2 , and Li_2O_2 .

Thanks you. We revised this sentence.

V) Line 157- Figure 3- the structure for the radical cation form of DMPZ is drawn incorrectly- one carbon drawn as a radical is implied to exceed the octet rule and another carbon has an implied hydrogen where none exists

Thank you for pointing us at this mistake, we corrected it.

W) Line 160- The caption sentence is written in such a way to imply that you are oxidizing the electrolyte solvent when you really mean to oxidize the DMPZ.

Thank you for pointing us at this, we corrected it.

X) Ilne 176 – the two abbreviatios are not consistent

a. Is B3LYP/6-31+G^* supposed to be the same as B3LYP/6-31G^*

Answer: No, they are different levels of theory used for optimizing geometries (B3LYP/6-31G^*) and a higher level of theory (B3LYP/6-31+G^*) for the energy calculations at the B3LYP/6-31G^* geometries. We have clarified the sentence so it will not be confusing:

Line 176: change to: The DFT energies of neutrals and cations were calculated at the B3LYP/6-31+G^* level of theory⁵⁸ for their geometries optimized at the B3LYP/6-31G^* level.

b. See also line 202 and line 180

Line 180, change to: In the case of the B3LYP/6-31+G^* energies

Line 202: This is consistent with line 176 because it is referring to the geometries that were optimized at the B3LYP/6-31G^* level.

Y) Line 189- the reference is not written correctly “(ref. 50)” should be “50”

According to journal style, where references are typically superscript numbers, references directly following a formula are written in the style (Ref. X)

Z) Line 231- since Table 1 was deleted why is it being referenced?

We corrected the mistake.

AA) Line 239-Line 252 The newly discussion section could use a little revision-> see comments in red

The results of this study have multiple implications for required research directions in Li-O₂ cells towards developing practical energy storage devices. First, generally, the previous paradigm that stability of cell components against superoxide [abbreviation?] and peroxide [abbreviation?] were of prime importance needs to shift towards additionally and even more importantly stability against 1O₂. This concerns both studies on how materials degrade and on making more stable materials. Second, redox mediation is now widely accepted to be key for Li-O₂ batteries to achieve maximum energy density and efficiency by far higher rates than possible without [this is rather awkward phrasing -what is meant by “higher rates” – a rate does not describe energy density or efficient?] The fact that we have shown that mediators can have very different susceptibility to decompose with 1O₂ spurs hope that even more stable mediators will be found. Third, the computational results that nicely reproduce the trend [this study only considers two different mediators!] in reactivity between different mediators and between the reduced and oxidized states suggest that computational screening is a very [since you didn't use computation to screen or preselect candidate mediator and you only investigates two mediator how can you make the claim that computational screening is an effective tool for screening---in fact you stated that you chose DMPZ for the low overpotential and TTF as one of the first reported RM's] effective tool to preselect candidate mediators. Fourth, even the best accessible [what is meant by accessible?] mediators may not be sufficiently stable for long term operation in presence of 1O₂. Therefore additional means of counteracting degradation will likely be required. These may be chemical traps that more rapidly react with 1O₂ than other cell components, or, preferably, physical quenchers.

Thank you for your kind suggestions. We revised the text accordingly. Regarding your comment on screening: it is right that we did not screen mediators here. But what we show is that the computational evaluation of the mediators reproduces the trend in reactivity found in our experiments. Therefore, the statement is meant to convey that if computational screening of other mediator candidates is done, it will predict reactivity trends correctly. We changed “screening is a very effective tool” to “screening will be a very effective tool”. WE changed “accessible” to “now available”.

BB) Line 252- what is physical quencher?- presumably you mean a molecule that can act to catalytically convert singlet oxygen to triplet oxygen- a little explanation here would help the reader

Yes, we mean this kind of molecules. We added the phrase “...physical quenchers which catalyze the decay from ¹O₂ to triplet oxygen⁴³.”

CC) Line 256- “Li “should read “lithium”

Thank you, we revised it.

DD) Line 259 – as previously mentioned, thermodynamically is not well defined in this statement--- see line 83 where a revision was made

We removed the phrase “thermodynamically and kinetically” since it was already previously defined what stable means.

Experimental

EE) Line 280- again what is a Swagelok design? your rebuttal gave a reference where presumably the construction of the cell is described but again the details of this cell are absent from your manuscript and cannot be found in the supplementary information

To clearly describe the “Swagelok cell” in our manuscript, we added the schematic image for cell structure in the new Supplementary Figure 7.

FF) Line 316- solvent peaks are typically used to reference spectra for example the chloroform peak in CDCl₃ is typically taken to be at 7.26 ppm. Solvent peaks are not used for quantification since the purity of the solvent peaks varies from bottle to bottle and will vary even between samples as it absorbs water from the environment.

We prepared all stability tests in parallel and furthermore prepared the samples for NMR spectroscopy with the very same deuterated solvent out of the same bottle (DMSO-d₆) within ~5 min. Hence, possible differences between bottles are excluded. Furthermore, DMSO is known to be very very mildly acidic. Considering that the pK_a values of water and DMSO differ by ~20 pK_a units, the exchange of protons of the water with deuterium atoms from DMSO-d₆ is extremely unlikely. For that reason, the relative amount of protons from DMPZ to DMSO can be used to relate the content of different samples. Knowing absolute values of undeuterated DMSO is not required. Hence, the values before and after photooxygenation can be used for quantitative comparison.

References

GG) Line 390- There are characters that do not appear to render correctly in the copy of the PDF that I have obtained. This may be a problem during publication. If it appears correct on your computer you may want to confirm with the editor that it will render correctly. Similar problems are seen at lines 390, 393, and 457.

Thanks for your careful check. We revised them.

HH) For your references “doi” links are given. Is this the correct format for this journal? I downloaded to recent nature communication papers and it does not seem that this is the case.

Thank you for pointing at this. It is correct at submission stage and will be changed during copy editing. However, as you commented out, we revised them.

II) Line 392- the phrase “doi” is repeated

We revised the references and removed “doi”.

JJ) Line 395- the phrase “doi” is repeated

We revised the references and removed “doi”.

KK) Line 397: you have the phrase “na/na” in the reference

Thank you, the mistake was corrected.

LL) Line 448: the phrase “doi” is repeated

We revised the references and removed “doi”.

Supplementary Information:

Line 37: is “photo-oxygenation” misspelled?

We corrected it.

Line 38: verb subject;” were” should be “was”

Thank you, the mistake was corrected.

Figure 4: what is the diameter of the cell?

Diameter of the cell (electrode) is 1 cm. This information was added in the caption of Supplementary Fig. 6 in the revised Supplementary Information.

REVIEWERS' COMMENTS:

Reviewer #1 (Remarks to the Author):

I am happy with the changes/corrections and believe that the manuscript can be published in its current form.